# Extracellular Matrix Proteomics: The *mdx-4cv* Mouse Diaphragm as a Surrogate for Studying Myofibrosis in Dystrophinopathy

**DOI:** 10.3390/biom13071108

**Published:** 2023-07-12

**Authors:** Paul Dowling, Stephen Gargan, Margit Zweyer, Dieter Swandulla, Kay Ohlendieck

**Affiliations:** 1Department of Biology, Maynooth University, National University of Ireland, W23 F2H6 Maynooth, Co. Kildare, Ireland; paul.dowling@mu.ie (P.D.); stephen.gargan@mu.ie (S.G.); 2Kathleen Lonsdale Institute for Human Health Research, Maynooth University, National University of Ireland, W23 F2H6 Maynooth, Co. Kildare, Ireland; 3Department of Neonatology and Paediatric Intensive Care, Children’s Hospital, German Center for Neurodegenerative Diseases, University of Bonn, D53127 Bonn, Germany; margit.zweyer@dzne.de; 4Institute of Physiology, Medical Faculty, University of Bonn, D53115 Bonn, Germany; swandulla@uni-bonn.de

**Keywords:** biglycan, collagen, dystrophin, dystrophinopathy, extracellular matrix, fibronectin, fibrosis, matrisome, mdx, periostin

## Abstract

The progressive degeneration of the skeletal musculature in Duchenne muscular dystrophy is accompanied by reactive myofibrosis, fat substitution, and chronic inflammation. Fibrotic changes and reduced tissue elasticity correlate with the loss in motor function in this X-chromosomal disorder. Thus, although dystrophinopathies are due to primary abnormalities in the *DMD* gene causing the almost-complete absence of the cytoskeletal Dp427-M isoform of dystrophin in voluntary muscles, the excessive accumulation of extracellular matrix proteins presents a key histopathological hallmark of muscular dystrophy. Animal model research has been instrumental in the characterization of dystrophic muscles and has contributed to a better understanding of the complex pathogenesis of dystrophinopathies, the discovery of new disease biomarkers, and the testing of novel therapeutic strategies. In this article, we review how mass-spectrometry-based proteomics can be used to study changes in key components of the endomysium, perimysium, and epimysium, such as collagens, proteoglycans, matricellular proteins, and adhesion receptors. The *mdx-4cv* mouse diaphragm displays severe myofibrosis, making it an ideal model system for large-scale surveys of systematic alterations in the matrisome of dystrophic fibers. Novel biomarkers of myofibrosis can now be tested for their appropriateness in the preclinical and clinical setting as diagnostic, pharmacodynamic, prognostic, and/or therapeutic monitoring indicators.

## 1. Introduction

Animal model research plays an important role in the field of biomedicine and, particularly, in biomolecular investigations into the basic mechanisms that underlie the cellular pathogenesis of common human diseases [1]. In the case of monogenetic disorders, such as the X-chromosomally inherited muscle-wasting disease named Duchenne muscular dystrophy [2], the animal models should ideally (i) mimic a similar genotype that properly reflects the primary abnormality leading to the complex disease phenotype; (ii) show comparable patterns of disease onset, progression, and severity; (iii) display comparable secondary changes as observed in the human disorder such as multi-system alterations and pathobiochemical crosstalk; (iv) be characterized by complex pathophysiological features that resemble the disease in patients including myonecrosis and myofibrosis; (v) exhibit equivalent effects due to changes in energy metabolism, cellular signaling pathways, physiological adaptations, and natural repair mechanisms; (vi) trigger analogous responses of the innate and/or adaptive immune system; (vii) be relatively cost-effective and easy to breed, handle, and house; (viii) be appropriate for efficient genetic manipulations; (ix) be suitable for standardized physiological, biochemical, cellular, or surgical manipulations, and (x) yield sufficient cellular and biofluid material for meaningful bioanalysis and high-throughput surveys [3,4,5]. Unless too many biological factors and mechanisms cause major differences in the pathophysiological progression between animal models and patients, then the results from studying disease models can at least partially be extrapolated for a better understanding of human disorders [6].

A variety of multifaceted skeletal-muscle-wasting diseases exist which are often characterized by a complex pathogenesis. Because of the high abundance of muscle fibers and their many interactions with other tissues, muscular disorders are often associated with body-wide changes and adaptations [7]. Since fibrosis is generally linked to chronic degeneration, oxidative stress, and inflammatory processes in human disorders [8,9], it is not surprising that myofibrosis is also seen in a variety of neuromuscular diseases [10]. Characteristic hallmarks of fibrotic changes in the skeletal musculature are the excess deposition of collagens and other extracellular matrix components [11]. Myofibrosis can be both of a primary or secondary nature due to reactive processes, such as the primary disorganization of the extracellular matrix in COL6-related myopathies [12,13,14] or reactive fibrosis as a secondary complication of dystrophinopathies [15,16,17], respectively.

X-linked muscular dystrophies that are based on primary abnormalities in the human *DMD* gene [2] and trigger the almost-complete absence of the full-length Dp427-M isoform of the cytoskeletal protein dystrophin [18] are highly progressive muscle-wasting diseases. Since the *DMD* gene contains several promoters that produce eight distinct and tissue-specific protein isoforms, different mutations affect individual patients in a disproportionate way. This is reflected by the dissimilar severity and onset of respiratory dysfunction, cardiomyopathy, kidney failure, liver disease, and abnormalities in the central nervous system [19]. Thus, the pattern of altered organ crosstalk, secondary abnormalities, and body-wide adaptations differ within patient cohorts. In the skeletal musculature, dystrophin is closely associated with a sarcolemmal glycoprotein complex, which is intrinsically involved in cellular signaling, the maintenance of plasmalemma integrity, the organization of the cytoskeleton, and the provision of lateral force transmission [20,21]. In the absence of the anchoring function of dystrophin, the protein complex consisting of dystroglycans, sarcoglycans, sarcospan, dystrobrevins, and syntrophins disintegrates and renders dystrophic fibers more susceptible to micro-rupturing. This causes an increased influx of Ca^2+^ ions into damaged muscle cells and promotes enhanced proteolysis and muscle necrosis [22,23,24]. The Ca^2+^-dependent damage pathway of myonecrosis is followed by fat substitution, chronic inflammation, and reactive myofibrosis [25,26,27].

This makes the development of novel anti-fibrotic strategies a crucial aim of muscle pharmacotherapy and applied myology in order to prevent and/or reverse fibrosis-related abnormalities that are linked to muscle dysfunction [28]. Animal model research plays a crucial role in the detailed analysis of the genetic, biochemical, and physiological parameters that underlie the precise mechanisms of fibrosis [29]. A critical aspect of this research focuses on how abnormal processes within the three main muscle-associated layers of connective tissue, i.e., the endomysium that surrounds contractile fibers, the perimysium that forms bundles around muscle fibers, and the epimysium that surrounds individual skeletal muscles [30,31,32], relate to chronic cellular damage, inflammatory responses, stem cell exhaustion, and alterations in the signaling pathways that regulate the fibro-adipogenic progenitor cell niche [33,34,35].

This article summarizes mass-spectrometry-based proteomic strategies to study the extracellular matrix in order to carry out unbiased investigations into the mechanisms that underlie skeletal muscle fibrosis. This includes an outline of the general importance and diverse utilization of the *mdx-4cv* mouse model in Duchenne muscular dystrophy research [36] with a special focus on fibrosis-related changes in the skeletal musculature as judged by mass spectrometric studies [37]. In contrast to the most widely used and naturally occurring *mdx-23* mouse, which is characterized by a point mutation in exon-23 of the murine *Dmd* gene and elevated levels of serum creatine kinase in association with histological lesions and cycles of muscle necrosis and regeneration [38,39,40], the *mdx-4cv* mouse has been generated by chemical mutagenesis [41]. This mouse model harbors a nonsense mutation that introduces a premature stop codon in exon-53 [42,43,44] and exhibits a considerably lower frequency of dystrophin-positive revertants [45] and different mutation-related alterations in the dystrophin isoform expression patterns [36] compared to the natural *mdx-23* mouse [40]. Dystrophin-lacking *mdx*-type mice exhibit varying degrees of fibrotic changes in the general musculature [46,47,48] but show severe reactive myofibrosis in the diaphragm muscle [49,50,51]. This histopathological hallmark in *mdx*-type diaphragm muscles is of considerable importance in the field of dystrophinopathy research [52,53], since endomysial fibrosis has been clearly established as a reliable marker that correlates well with motor loss in Duchenne patients [54]. The aged *mdx-4cv* mouse diaphragm, especially, shows high levels of extracellular matrix deposition [55], making it a suitable surrogate for studying reactive myofibrosis in dystrophinopaty, as discussed in this review.

## 2. Proteomic Strategies to Study the Extracellular Matrix

Mass spectrometry has been instrumental for the systematic cataloguing and characterization of the human proteome [56,57,58,59,60]. A large number of studies have focused on the establishment of the skeletal muscle proteome with over 10,000 identified protein species in contractile tissue specimens from both humans and a variety of animal species that are frequently used in biomedical research [61,62,63,64,65,66]. Since the extracellular matrix plays a key role in tissue organization, cellular integrity, wound healing, and repair mechanisms, proteomics has been extensively applied to characterize the matrisome and its associated proteins in many cell/tissue/organ types and species [67,68,69]. Cell biological studies in combination with advanced biochemical separation approaches and systematic protein identification by mass spectrometry have been instrumental for the general proteomic cataloguing of the overarching extracellular matrix system that is present in many different cell and tissue types [70,71,72,73]. The bioanalytical advantages versus technical limitations of the main methodological strategies and proteomic analysis pipelines that are routinely used for the characterization of the subproteome that is associated with the extracellular matrix have been outlined in previous reviews [74,75,76]. Figure 1, Figure 2 and Figure 3 summarize (i) direct and extracellular-matrix-enrichment-independent preparations for the bottom-up proteomic profiling of the skeletal muscle proteome including the accessible matrisome, (ii) proteoform-centric and extracellular-matrix-enrichment-independent preparations for the top-down proteomic profiling of the skeletal muscle proteome including the accessible matrisome, and (iii) extracellular-matrix-enrichment-based proteomic-profiling strategies for studying the matrisome and using sequential depletion or decellularization methodology, respectively.

### 2.1. Bottom-Up Proteomics versus Top-Down Proteomics

In principle, the main proteomic approaches used in the biological sciences can be categorized into two main types, bottom-up proteomics and top-down proteomics [77]. Bottom-up proteomics focuses on the large-scale identification of proteins using crude extracts as the starting material and peptide mass spectrometry for protein detection [78]. Top-down proteomics is usually based on the biochemical or gel electrophoretic separation of intact protein species which is then followed by the mass spectrometric analysis of intact proteoforms and their post-translational modifications [79]. An overview of the main techniques employed in skeletal muscle proteomics, including a description of the main analysis platforms, two-dimensional gel electrophoresis, liquid chromatography, antibody-based approaches, sample preparation, protein digestion protocols, mass spectrometry, data acquisition, single-cell analysis, and aptamer-based proteomics, has recently been published in the context of the proteomic profiling of the contractile apparatus of skeletal muscles [80]. Of note, single-cell proteomics is becoming increasingly important [81] and has been applied for studying individual muscle fibers [64,65]. Studying changes in specialized muscle fibers is crucial since fiber type shifting occurs in the skeletal musculature during physiological adaptations and in many neuromuscular disorders [82]. Figure 1 and Figure 2 provide an overview of commonly used methods for the bottom-up or top-down proteomic analysis of crude protein extracts from total muscle tissue specimens without the prior usage of extensive enrichment processes [79,83].

### 2.2. Sample Handling and Protein Extraction for Gel-Free and Bottom-Up Proteomics

As summarized in Figure 1, efficient protein extraction for bottom-up muscle proteomics can be performed by various direct sample preparation techniques [83,84,85], such as the widely used detergent-based and filter-aided sample preparation (FASP) method [86,87], enhanced eFASP using alternative detergents [88], sample processing from cell lysis through the elution of purified peptides by the In-StageTip (iST) method [89,90], the single-pot solid-phase-enhanced sample preparation (SP3) method [91], the universal solid-phase protein preparation (USP3) method [92], pressure-cycling technology (PCT) [93], surfactant-and-chaotropic-agent-assisted sequential extraction/on-pellet digestion (SCAD) [94], or by detergent-free sample preparation by easy extraction and digestion (SPEED) [95]. A comprehensive description of a standardized bottom-up proteomic workflow, incorporating the FASP method, has recently been described in detail [96].

### 2.3. Sample Handling and Protein Extraction for Gel-Based and Top-Down Proteomics

Although the majority of mass spectrometric surveys are carried out with peptides and proteins that are isolated by gel-free systems, top-down proteomics routinely employs one-dimensional or two-dimensional gel electrophoretic techniques, or biochemical isolation methods such as affinity chromatography, for the preparation of intact proteoforms. As summarized in Figure 2, gel-based top-down proteomics can be based on GeLC-MS/MS using one-dimensional gradient gels in combination with liquid chromatography [97,98,99] or classical two-dimensional gel electrophoresis using isoelectric focusing in the first dimension and slab gel electrophoresis in the second dimension [100,101,102]. A gel-based method that has overcome the technical limitation of gel-to-gel variations is represented by two-dimensional fluorescence difference gel electrophoresis (2D-DIGE) [103,104,105]. This advanced gel technique employs differential proteome tagging with the CyDyes Cy2, Cy3, and Cy5 in combination with various software analysis packages to determine protein changes in comparative proteomics [79].

### 2.4. Extracellular-Matrix-Enrichment-Based Proteomic-Profiling Approaches

The main enrichment strategies for mass spectrometric studies of the isolated extracellular matrix are summarized in Figure 3, including both sequential depletion methods and a list of decellularization methods. The matrisome can be divided into components that belong to the insoluble versus the soluble extracellular matrix, often referred to as the iECM versus the sECM [74,75,76]. Sequential depletion approaches focus on the isolation of an insoluble pellet consisting of key extracellular matrix proteins via the systematic removal of soluble protein species by various washing steps. Commercially available ‘Compartmental Protein Extraction’ kits represent a convenient way to carry out the sequential solubilization and extraction of major subcellular fractions, such as the cytosol, nucleus, membranes, and cytoskeletal networks, and thereby enrich the iECM part of the matrisome as an insoluble pellet [75,106]. In contrast, the sECM can be prepared by a ‘high salt’ method that aims to isolate the chaotropic-agent-soluble matrisome. High-salt buffers are employed to remove the soluble protein fraction, followed by extracellular matrix extraction using chaotropic reagents [75,107]. Quantitative detergent solubility profiling (QDSP) is based on the sequential treatment with buffers of increasing detergent and salt concentration and harvesting of the matrisome as an insoluble pellet [75,108]. An interesting approach is the usage of the photo-cleavable surfactant 4-hexylphenylazosulfonate (Azo). Following treatment with the nonionic detergent Triton X-100 to produce an insoluble pellet, the Azo detergent is then used for the specific solubilization and isolation of the extracellular matrix fraction in association with the ultraviolet irradiation of Azo [75,109,110]. The ‘three-step extraction’ method is a sequential extraction process that is based on a high-salt step, a detergent solubilization step, and a final chaotropic-agent-based solubilization of the matrisome fraction [75,111].

Decellularization methods to treat tissue samples usually start with the disruption and removal of cellular material using various biochemical, chemical, or physical techniques, followed then by the isolation of the mostly insoluble extracellular matrix fraction [74,75,76]. Freeze–thaw cycles, agitation, pressurization, sonication, and osmotic pressure are typical methods used for cellular disruption. Biochemical approaches used for decellularization often use chemicals such as SDS, Triton X-100, and CHAPS, followed by the treatment of the insoluble matrisome with chaotropic agents, including high concentrations of urea, thiourea, or guanidine hydrochloride [111,112,113]. Prior to peptide mass spectrometry, the enriched extracellular matrix fraction is often digested by trypsin and LysC, but also with the help of cyanogen bromide or hydroxylamine [74,75,76].

### 2.5. Mass Spectrometry and Data Acquisition

The identification and characterization of individual proteoforms in crude tissue samples, subcellular fractions, enriched preparations of the matrisome, or isolated protein assemblies can be carried out by a variety of mass spectrometric methods [114,115,116,117], such as liquid chromatography–tandem mass spectrometry (LC-MS/MS) [115,118] or matrix-assisted laser desorption/ionization time-of-flight mass spectrometry (MALDI-TOF MS) [114,119]. Both labelling and label-free approaches are routinely used for the detection of protein species. The bioanalytical advantages versus technical limitation of these frequently used methods have been reviewed in detail [120]. Stable heavy isotopes can be incorporated into peptides/proteins using chemical or metabolic labelling and then be used for the quantitation of protein abundance [121]. Examples of well-established proteomic-labelling methods are stable isotope labelling by amino acids in cell culture (SILAC) [122], the tandem mass tagging (TMT) technique [123], the isotope-coded affinity tagging (ICAT) approach [124], and the isobaric tagging for relative and absolute quantitation (iTRAQ) technique [125]. Alternatively, mass spectrometric analyses can be performed with label-free quantitation techniques that focus on features from fragment-ion analysis (MS2), such as spectral counting, or measure the integrated peak intensity from the parent-ion mass analysis (MS1) [126,127]. Of note, the application of artificial intelligence (AI), machine learning (ML), and deep learning (DL) algorithms has been increasingly used to evaluate complex spectral data [128].

Optimized data acquisition plays a key role in mass spectrometric analyses. This can be achieved by data-dependent acquisition (DDA) [129], data-independent acquisition (DIA) [130], or targeted data acquisition (TDA) [131]. In the context of DIA, the sequential window acquisition of all theoretical mass spectra mass spectrometry (SWATH-MS) method, which divides the mass range into small mass windows, has been successfully applied to studying extracellular matrix proteins [132]. Immunoprecipitation combined with SWATH-MS was successfully employed to investigate fukutin-related protein (FKRP)-associated dystroglycanopathy [133]. This methodology revealed that the loss of FKRP function disrupts the localization and tethering of fibronectin to myosin-10, essential for fibronectin maturation and subsequent collagen binding. The evidence accumulated demonstrates the utility of combining SWATH-MS and enrichment methods to investigate distinct phenotypes associated with muscular disorders.

Important methods for focusing on the abundance of target peptides include selected/multiple reaction monitoring (SRM/MRM) [134] or parallel reaction monitoring (PRM) [135], which are based on the measurement of precursor peptides in predefined *m*/*z* ranges taking into account retention time. The MRM-based measurement of targeted peptides was recently applied for the detailed analysis of extracellular matrix proteins [136]. Importantly, the absolute quantification (AQUA) workflow involves the usage of stable-isotope-labelled peptides that are spiked into the samples at predetermined concentrations and are then quantified simultaneously with the corresponding endogenous peptides. This greatly facilitates better accuracy, sensitivity, and reproducibility of protein detection [137].

## 3. The Extracellular Matrix of Skeletal Muscles

The extracellular matrix of skeletal muscle fibers plays a key role in the provision of force transmission and the external support for the continued stabilization, maintenance, and repair of contractile fibers [30,31,32], which is reflected by complex structure–function relationships [138,139]. Importantly, the processes that regulate and support cell adhesion and cell migration during development, wound healing, and regeneration involve extensive cell–matrix interactions, which warrants detailed omics-type studies into the composition of the matrisome. In the field of proteomics including extracellular matrix proteomics, it is important to stress the now generally accepted concept of the proteoform being the basic unit of the dynamic proteome [140,141].

Thus, the detailed biochemical information on individual proteoforms and their post-translational modifications (PTMs), as detected by mass spectrometry, should be used to establish detailed proteomic maps of the matrisome. Importantly, PTM profiles of extracellular matrix proteins can provide important biochemical characteristics for detecting and understanding the complex pathways associated with skeletal muscle physiology and pathophysiology. Mounting evidence exists showing how the existence of various PTMs are essential for matrisomal protein secretion and functionality within the extracellular milieu. The thrombospondin type-1 domain (TSR) is a common molecular feature present in the thrombospondin family, where O-fucosylation has a crucial stabilizing effect on the TSR structure [142]. Methods for detecting hydroxylation and glycosylation sites in collagen IV have previously been documented [143]. Such methods are important steps for understanding the role of specific PTMs in health and disease, as hydroxylation is essential for the stability of collagen molecules, and associated fibril structures. Disulfide bonds, assembled through the action of protein disulfide isomerases (PDIs), are found in thrombospondins and fibronectin, playing important roles in the establishment of subunit stability [144]. Citrullination, in which the conversion of the amino acid arginine into the amino acid citrulline is established, is associated with several extracellular matrix proteins, including fibronectin, having important roles in protein clustering, focal adhesion stability, and signaling [145].

Table 1 lists the representative protein hit list using the standardized mass spectrometric identification of proteins belonging to the various layers of the extracellular matrix from wild-type mouse diaphragm muscle [37,146]. This proteomic analysis of matrisomal proteins was carried out by bottom-up proteomics [96] incorporating the detergent-based and filter-aided sample preparation (FASP) method [86,87]. The proteins listed in Table 1 cover proteoforms that are found throughout the endomysium, perimysium, and epimysium, as well as protein species that are enriched in the basal lamina, myotendinous junctions, tendons, and cartilage [147]. Major members of the proteoglycan family, matricellular proteins, and adhesion receptors of the sarcolemma were clearly identified by proteomics [17,37,146].

Load bearing, fiber stretching, and continuous cycles of excitation, contraction, and relaxation put enormous physical strain on individual skeletal muscles. The surrounding layers of the extracellular matrix protect the muscular system from excess damage. The physical scaffolds provided by collagens and proteoglycans act as a supporting structure that embeds individual motor units with their motor neurons and corresponding muscle fibers, as well as capillaries and satellite cells [30,31,32]. Of special importance for the integrity of the sarcolemma is the linkage between the intracellular cytoskeleton and the extracellular basement membrane. Listed in Table 1 as a key component of the basal lamina is the heterotrimeric glycoprotein named laminin-211 [147], a multi-functional member of the large laminin family of basement membrane proteins [148]. Laminin-211, whose name refers to its composition consisting of the merosin alpha-2 (LAMA2) subunit, the beta-1 (LAMB1) subunit, and the gamma-1 (LAMC1) subunit, was shown to be involved in fiber stabilization and cell adhesion, as well as cell differentiation and proliferation mechanisms [148]. Laminin-211 can interact with collagens, integrins, the dystroglycan complex, nidogen, and heparan sulfate at the level of the plasma membrane [149] and also plays a central role in the organization of the neuromuscular junction [150].

An important binding partner for the laminin complex is the heparan sulfate proteoglycan named agrin, which mediates the aggregation of the nicotinic acetylcholine receptors at the post-synaptic structures during synaptogenesis [151]. This extracellular matrix component of the neuromuscular junction was identified by mass spectrometry in crude muscle extracts (accession number A2ASQ1-3; *Agrn* gene; 2 peptides; 1.6% sequence coverage; 198.2 kDa molecular mass) despite its low abundance. Additional basal-lamina-enriched extracellular matrix proteins were detected as collagen isoforms IV (COL4A1), XV (COL15A1), and XVIII (COL18A1) [152], perlecan (HSPG-2) [153], and the nidogens named entactin (NID1) [154] and osteonidogen (NID2) [155], as listed in Table 1. Collagen IV is a major member of the muscle collagen family [156] and is highly enriched in the basal lamina [157]. Perlecan of 398 kDa is an abundant component of the basal lamina and also known as the basement-membrane-specific heparan sulfate proteoglycan core protein (HSPG-2) [153]. The nidogens are intrinsically involved in the assembly of the basal lamina and provide a linkage between the laminin complexes and the collagen IV system, as well as perlecan in the basement membrane [156,157].

Endomysium-enriched extracellular matrix proteins are represented by specific collagens, as listed in Table 1, including collagen V (Col5a1) and collagen VI (Col6a1, Col6a2, Col6a5, Col6a6) [158]. The collagen lattice and associated proteoglycans form an intricate extracellular maintenance system that supports muscle tissue integrity and fiber elasticity, as well as binding sites for trans-plasmalemmal linkage to the cytoskeletal networks to stabilize excitation–contraction–relaxation cycles. Of note, collagen VI plays a key role in the structural stabilization of the neuromuscular junction [159]. Via costameres, the extracellular matrix mediates the transduction of muscle force towards the surrounding tissues. Extracellular matrix components support crucial signaling pathways and thereby preserve neuromuscular homeostasis [160]. Extracellular matrix proteins that can be found throughout the endomysium, perimysium, and epimysium include prolargin (PRELP), dermatopontin (DPT), vitronectin (VTN), fibronectin (FN1), fibrinogen (FGA, FGB, FGG), the microfibril-associated glycoprotein MFAP-4, and fibrillin (FBN1) [32,138]. The large group of muscle-associated small leucine-rich proteoglycans (SLRP type) was represented by asporin (ASPN), biglycan (BGN), decorin (DCN), fibromodulin (FMOD), lumican (LUM), and mimecan/osteoglycin (OGN) [32,161]. As listed in Table 1, mass spectrometry detected key members of the family of non-structural matricellular proteins, including periostin (POSTN), thrombospondin (THBS1, THBS4), and fibulin isoform FBLN5 [162,163].

Major sarcolemmal adhesion proteins that provide cell–matrix linkages are the alpha-7/beta-1 integrin (ITGA7, ITGB1) complex and the dystroglycan (alpha/beta-DAG1) complex, which are both involved in force transmission and signaling mechanisms. See Table 1 for the proteomic identification of these sarcolemmal protein receptors. The alpha-7/beta-1 integrins form an abundant trans-sarcolemmal adhesion site at costameres and myotendious junctions that links laminin and fibronectin of the extracellular matrix to the intracellular cytoskeleton [164]. The dystroglycans form the core of the sarcolemmal dystrophin–glycoprotein complex and mediate the linkage between laminin and cortical actin [165,166,167] via tight interactions between the laminin-binding protein alpha-dystroglycan [168], the integral glycoprotein beta-dystroglycan [169], and the actin-binding domain of the Dp427-M isoform of the membrane cytoskeletal protein dystrophin [20]. The organization of the various layers of the extracellular matrix and the distribution of individual protein components is diagrammatically summarized in Figure 4.

Myotendinous-junction-enriched proteins listed in Table 1 are collagen XII (COL12A1) and FERM-domain-containing kindlin 2 (FERMT-2) [170,171,172], and the abundant collagens of the tendon are represented by the microfibrillar isoform collagen I (COL1A1, COL1A2) [173]. The group of cartilage-enriched proteins was detected in the form of the cartilage oligomeric matrix protein COMP and the cartilage intermediate layer protein (CILP, CILP2). Extracellular matrix proteins involved in cellular signaling are the transforming growth factor-beta-induced protein ig-h3 (TGFBI) and protein-glutamine gamma-glutamyltransferase TGM2. Extracellular-matrix-associated proteins that belong to the annexin family of proteins were identified as the annexin isoforms ANXA2 and ANXA6, whereby Annexin-6 is a key protein involved in membrane repair [174]. Extracellular matrix regulators belonging to the serpin family include the alpha-antitrypsins A1AT4 (SERPINA1D), A1AT2 (SERPINA1B), A1AT3 (SERPINA1C), and A1AT5 (SERPINA1E), as well as Serpin-H1 (SERPINH1), serine protease inhibitor A3K (SERPINA3K), and cathepsin-B (CTSB) [175,176].

The above-listed components of the extracellular matrix form a highly adaptable, dynamic, and acellular microenvironmental system that supports, embeds, and stabilizes skeletal muscle fibers and their physiological functions [177]. The complex meshwork of extracellular laminins, non-fibrillar collagens, collagen microfibrils, proteoglycans, matricellular proteins, and regulating enzymes, and their linkage to cell-associated adhesion receptors in the sarcolemma membrane, is involved in large numbers of cell biological processes in contractile tissues, including:The stabilization of skeletal muscle fibers, and the provision of tissue strength and elasticity;Structural support via physical scaffolding, and the provision of an embedding medium for motor units, capillaries, and pools of satellite cells;The provision of neuromuscular homeostasis;Mechanical force transduction from muscle fibers to the surrounding tissues;The provision of cell–matrix interactions for the support of cell adhesion and cell migration during embryonic myogenesis, adult myogenesis, and tissue repair;Cell–matrix support during cell differentiation, maturation, remodeling, fiber transitions, and muscle aging.

The bioinformatic STRING analysis [178] of the muscle matrisome is shown in Figure 5, which was carried out with the proteins listed in Table 1 above. The interaction map illustrates the close protein–protein linkage patterns within the extracellular matrix of skeletal muscles.

## 4. The *mdx-4cv* Mouse Model and Dystrophinopathy-Associated Myofibrosis

### 4.1. Duchenne Muscular Dystrophy and Fibrosis

The most severe form of dystrophinopathy, Duchenne muscular dystrophy, is a primary-muscle-wasting disorder of early childhood [179] with a prevalence of approximately 1 in 5000 live male births [180], which is triggered by a variety of genetic abnormalities in the X-chromosomal *DMD* gene [181]. Initial clinical signs include developmental delays in Duchenne patients with proximal muscle weakness, temporal and spatial gait variations, decreased walking speed, and Gower’s sign [182,183,184]. Characteristic patterns of toe walking and difficulties with climbing stairs progress at more advanced stages of the disease towards respiratory insufficiency, cardiomyopathy, scoliosis, and severe limitations in general mobility [185,186,187,188,189,190], which eventually results in the loss of unassisted ambulation and upper body weakness [191]. Approximately one-fifth of Duchenne patients suffer from intellectual developmental disorder that can be associated with learning difficulties, behavioral deficits, emotional problems, attention deficits, impaired language and speech development, cognitive deficiencies, and mental retardation [192,193,194]. Of note, secondary multi-system abnormalities in dystrophinopathy negatively affect whole-body homeostasis, including the proper functioning of the kidneys, the bladder, the gastrointestinal tract, and the liver [19].

Detailed descriptions of crucial aspects of dystrophinopathy can be found in extensive review articles that focus on the epidemiology [180], X-linked genetics [2], cellular pathogenesis [179], differential diagnosis [195], rehabilitation strategies [196], and emergency management [197] of Duchenne muscular dystrophy. Following the identification of the *DMD* gene and its involvement in X-linked Duchenne muscular dystrophy [198,199,200] and the discovery of the full-length muscle dystrophin protein (Dp427-M) and its many isoforms (Dp427-B, Dp427-P, Dp260-R, Dp140-B/K, Dp116-S, Dp71, and Dp45) [18,201,202,203], detailed biochemical, biophysical, and physiological studies have established that full-length dystrophin acts as a sub-sarcolemmal anchoring node that is involved in cytoskeletal organization, the provision of lateral force transmission, the maintenance of muscle fiber stability, and the support and integration of cellular signaling mechanisms in skeletal muscles [20]. The loss of the dystrophin isoform Dp427-M destabilizes the entire dystrophin-associated glycoprotein complex and causes a drastic reduction in the core dystroglycan sub-complex at the sarcolemma membrane, as well as lower levels of sarcoglycans, sarcospan, dystrobrevins, and syntrophins [204]. The dystrophin-deficient muscle membrane system was shown to be more prone to contraction-induced rupturing and this, in turn, results in a pathophysiological Ca^2+^ influx into muscle fibers. The chronically elevated Ca^2+^ concentration increases the rate of calpain-mediated proteolysis and tissue destruction [22,23,24].

Dystrophic skeletal muscles are characterized by highly progressive myonecrosis [19] and chronic inflammation [27], which is accompanied by reactive myofibrosis [17,26,37,50]. On the histological level, the drastic decline in muscle strength is characterized by fiber branching, alterations in fiber diameter, roundly shaped myofibers, hypercontractility, a high level of central myonucleation, intra-muscular fatty accumulation, muscle infiltration by inflammatory cells, and high levels of collagen deposition. In general, muscle-degeneration-induced myofibrosis is usually characterized by increased mechanical stiffness in contractile fibers with extensive levels of fibrillar disorganization and drastically increased deposition of collagens, as well as high levels of myofibroblasts and macrophages [25,26,27,205]. The histopathological changes that result in endomysial fibrosis reflect these changes and they correlate with the loss in motor function in Duchenne patients [54], making the degree of reactive myofibrosis a useful indicator of disease progression in dystrophinopathy [17]. A variety of muscle-associated cell types, including resident fibroblasts, the important category of fibro-adipogenic progenitors, inflammatory cells, and pericytes, can be activated to transform towards the myofibroblast phenotype [33,34,35]. This complex network of activated cells and drastically increased deposition of extracellular matrix components reflects the pathophysiological impact of reactive myofibrosis on the cellular pathogenesis of Duchenne muscular dystrophy [11]. This gives the development of novel anti-fibrotic pharmacotherapies a central position in the design of new treatment options to counteract dystrophinopathy-associated muscle pathology [28,206,207,208,209].

### 4.2. Characterization of the mdx-4cv Mouse within the Context of Other Dystrophic Models

Both spontaneous and bioengineered mouse models that lack the dystrophin protein isoform Dp427-M have been instrumental in the detailed elucidation of the molecular and cellular pathogenesis of Duchenne muscular dystrophy, as well as the evaluation of experimental treatment strategies [210,211,212]. A comparative listing of major genetic mouse models of dystrophinopathy is provided in Table 2. This includes the original *mdx-23* mouse compared to a variety of modified models including *mdx/Dtna*, *mdx/Cmah, mdx/Utr*, *mdx/α7*, *mdx/Myod1*, *mdx-2cv*, *mdx-3cv*, *mdx-4cv*, *mdx-4cv/mTR-G2*, *mdx-5cv*, *mdx-52*, *mdx-βgeo*, *Dmd-null*, *hDMD/mdx-45,* and *hDMD/mdx-52*. The table lists information on the genetic abnormalities that characterize the various mouse models and the observed severity of the dystrophic phenotype in individual models of Duchenne muscular dystrophy. Importantly, Table 2 gives an overview of the effect of mutations on the expression of distinct dystrophin isoforms ranging from Dp71 to Dp427. Depending on the genetic abnormality within the *Dmd* gene, impaired expression patterns might only cause the loss of the full-length dystrophins Dp427-M, Dp427-B, and Dp427-P, or also additional effects on the tissue-specific density of the shorter dystrophins Dp260-R, Dp140-B/K, Dp116-S, Dp71, and/or Dp45. Detailed comparisons of dystrophic mouse models have been carried out in recent reviews [213,214,215].

Compared to the onset of dystrophic changes in the skeletal musculature at approximately 1 month of age in the *mdx-23* mouse and a maximum lifespan of approximately 2 years, the modified *mdx*-type mice differ considerably in the onset of histopathological changes, ranging from 2 to 8 weeks, and their longevity, ranging in lifespan from 1 to 23 months [216]. Both the spontaneously mutated *mdx-23* mouse and the chemically mutated *mdx-4cv* mouse have comparable survival rates and similar onsets of moderate hindlimb degeneration, as judged by histopathology, at 3–4 weeks of age in association with physiological dysfunction such as reduced muscle force. The *mdx-23* mouse is characterized by extensive cycles of degeneration and regeneration from 3 to 8 weeks, illustrated by high levels of central myonucleation, and some stabilization with age [38,39,40]. The *mdx-4cv* musculature shows large variation in skeletal muscle fiber sizes. In both model systems, the dystrophin-deficient diaphragm is severely affected starting at 3–4 weeks of age and shows high levels of reactive myofibrosis during aging [213,214,215]. Analogous to the skeletal musculature, the reduced presence of the cardiac dystrophin–glycoprotein complex causes sarcolemmal disintegration, myonecrosis, fatty tissue replacement, fibrotic scarring, and interstitial inflammation in the *mdx*-type heart [217]. The onset of cardiomyopathy is at approximately 6–9 months of age and associated with considerable proteome-wide changes during aging [218].

The range of modified *mdx*-type mouse models of dystrophinopathy with usually more severe phenotypes includes double mutants that also affect, besides the dystrophin levels, the expression of proteins such as the dystrophin-associated component dystrobrevin, the autosomal dystrophin homologue utrophin, myogenic differentiation protein MOD1, and α7-integrin [213,214,215,216]. In addition to the most frequently used murine models with their rapid disease progression, large animal models of Duchenne muscular dystrophy exist in the form of canine and porcine mutants [5,219,220,221]. The Golden Retriever muscular dystrophy (GRMD) model exhibits a splice site mutation in intron-6, which causes the skipping of exon-7 and a resulting out-of-frame *DMD* transcript in exon-8 [220]. The complexity and progression of the clinical syndrome is severe in the GRMD model including cardiomyopathic complications, making dystrophic dogs good model systems for translational studies [222,223]. The porcine model of dystrophinopathy (DMD pig) is characterized by a deletion in exon-52 of the *DMD* gene and exhibits progressive cardiomyopathy [224]. In conjunction with mouse models, larger animals can be utilized to study pathophysiological mechanisms, validate the suitability and robustness of novel biomarker candidates, and be helpful during the preclinical testing phase of new therapeutic approaches such as gene editing [219,221]. Large disease-model animals are highly useful for facilitating the efficient translation of novel diagnostic methods to the clinical setting, such as imaging technology for the monitoring of myofibrosis [225].

Skeletal muscle fibrosis is seen in different *mdx*-type muscles to varying degrees [46,47,48,226], with the diaphragm being the most structurally affected and fibrotic muscle tissue [49,50,51,52,227,228]. However, one of the most widely employed dystrophic animal models, the *mdx-23* mouse [38], exhibits a relatively high frequency of revertant fibers [39] and a relatively mild dystrophic phenotype [40]. In contrast, the chemically mutated *mdx-4cv* mouse which harbors a nonsense mutation in exon-53 [41,42,43] has a drastically reduced number of dystrophin-positive revertants [45]. Figure 6 provides a comparison of the main genetic, biochemical, physiological, and histological features of muscular abnormalities in Duchenne muscular dystrophy versus the widely used *mdx*-23 and *mdx-4cv* mouse models of dystrophinopathy.

Although the *mdx-4cv* mouse with a nonsense point mutation that introduces a premature stop codon in exon-53 [41] does not reflect the genetic heterogeneity seen in large cohorts of human Duchenne patients [2], consisting of a variety of splice site mutations, missense point mutations, nonsense point mutations, large deletions, large duplications, small deletions, small duplications, and mid-intronic mutations [181], the pathobiochemical down-stream effects due to dystrophin deficiency are relatively comparable. The almost-complete absence of the Dp427-M isoform of full-length dystrophin in skeletal muscle fibers initiates the reduced anchoring of the alpha/beta-dystroglycan subcomplex, alpha/beta/gamma/delta-sarcoglycans, sarcospan, syntrophins, and dystrobrevins [146,229,230,231]. The collapse of the dystrophin-associated glycoprotein complex renders the muscle plasma membrane more susceptible to contraction-induced rupturing followed by the chronic influx of Ca^2+^ ions both through the leaky sarcolemma and Ca^2+^-leak channels. The resulting elevation of sarcosolic Ca^2+^ levels and concomitant impairment of the luminal Ca^2+^-buffering capacity of the sarcoplasmic reticulum negatively affect the fine regulation of the excitation–contraction coupling mechanism and overwhelm the Ca^2+^-extrusion system of damaged muscle fibers resulting in Ca^2+^-dependent proteolysis [22,23,24]. The histopathological consequences of the interplay between progressive fiber degeneration and abnormal ion homeostasis causes increased levels of inflammatory cells, impaired myoblast cell survival, central nucleation in mature fibers due to ongoing cycles of degeneration and regeneration, and increased levels of extracellular proteins [52,229,232]. One histopathological feature that is not observed in *mdx-4cv* muscles, compared to dystrophic skeletal muscles from both Duchenne and Becker’s muscular dystrophy patients [233,234,235,236], are high levels of fat substitution.

However, progressive myonecrosis, chronic inflammation, and reactive myofibrosis with increased levels of collagens, proteoglycans, matricellular proteins, and annexins are clearly observed in the *mdx-4cv* model with increasing severity during aging [55]. Table 3 lists major studies that have been carried out to characterize the *mdx-4cv* mouse model in the context of (i) chemical mutagenesis, genotyping, and phenotyping [41,42,43,44,231,237]; (ii) the testing of novel therapeutic strategies to treat X-linked Duchenne muscular dystrophy, including exon-skipping therapy [238], virus- or nanocarrier-mediated micro-dystrophin gene therapy [239,240,241,242,243], cell-based therapy [244,245,246], and gene editing [247,248]; (iii) the biochemical profiling of proteome-wide changes due to multi-systemic changes, including various skeletal muscles [55,146,229,230,249,250,251,252], heart [253], liver [254], kidney [255,256], spleen [96,257], stomach wall and pancreas [258], and central nervous system [259]; and (iv) the biochemical profiling of proteome-wide changes in biofluids including serum [260,261], urine [262], and saliva [261,263]. The *mdx-4cv* mouse has also been used to produce an immunodeficient muscular dystrophy model, *NSG-mdx-4cv* [264,265,266,267], and an *mdx-4cv/mTR-G2* model with humanized telomere lengths [216,268], as listed in above Table 2. The testing of experimental therapies with suitable animal models is crucial for the establishment of novel therapeutic options to treat Duchenne muscular dystrophy [5,215,269], such as cell transfer therapy, genomic editing, exon-skipping therapy, and multi-drug combination pharmacotherapy [270,271,272,273,274,275], including anti-fibrosis strategies [28].

### 4.3. Histological and Biochemical Characterization of Fibrosis in the mdx-4cv Mouse Diaphragm

The diaphragm muscle is an important type of tissue for the focused analysis of pathophysiological changes since it is majorly affected in Duchenne patients that suffer from impaired cardiorespiratory function. A reduced thoracic cavity area and altered chest wall contraction patterns have been established to occur during cycles of inspiration and expiration in dystrophic respiratory muscles by imaging studies [276]. Severe diaphragmatic dysfunction, excessive fat infiltration in respiratory muscles, and cardiomyopathic complications occur mostly in the second decade of life in inherited muscular dystrophy, which frequently necessitates both the management of respiratory insufficiency by mechanical ventilatory support [277] and targeted drug therapy of the weakened and dystrophin-deficient heart muscle [185,278]. However, respiratory decline can already be observed during the early ambulatory phase in dystrophic boys [279] with diaphragmatic motions being drastically impaired following inspiration [280]. Analogous to the severely degenerative muscle phenotype seen in Duchenne patients, the aged *mdx-4cv* mouse diaphragm muscle shows the main histopathological hallmarks of dystrophinopathy [49,52]. This is summarized in the histological and immunofluorescence microscopical analysis depicted in Figure 7.

The illustration highlights myonecrosis, reactive myofibrosis, and zones of chronic inflammation. The dystrophic diaphragm exhibits more roundly shaped myofibers and a high frequency of central myo-nucleation, which is due to ongoing cycles of fiber degeneration and regeneration. These histopathological changes clearly correlate with skeletal muscle dysfunction and severe force deficits in the dystrophic *mdx*-type diaphragm [281]. However, one additional sign of muscular changes, which is often observed in human patients, is not present in dystrophic *mdx-4cv* mice, i.e., fat substitution. Immuno-labelling of dystrophin isoform Dp427-M confirmed the sarcolemmal localization of this membrane cytoskeletal protein in wild-type diaphragm and its almost-complete loss in the dystrophic and highly fibrotic *mdx-4cv* diaphragm. To complement microscopical studies, a comparative immunoblot analysis is routinely used to independently verify findings from proteomic studies. Western blotting surveys have clearly confirmed the drastic increase in many extracellular matrix proteins in muscular dystrophy, including various fibrosis markers such as collagens, fibronectin, dermatopontin, biglycan, and periostin in *mdx*-type muscles [282,283,284]. The glycoprotein fibronectin [285] is an exciting fibrosis-related biomarker candidate since its levels are also significantly increased in the serum of Duchenne patients compared to age-matched controls [286]. The immunoblots shown in Figure 8 illustrate the elevated levels of collagen isoform COL-VI, which, in muscle tissue, consists of three alpha chains [158], the annexin isoform ANXA2, and the matricellular protein periostin in the *mdx-4cv* diaphragm. It is not surprising that enhanced collagen deposition and a high concentration of fibronectin in fibrotic muscle tissue are closely related to increased dermatopontin levels since this tyrosine-rich acidic matrix protein plays a central role in cell–matrix interactions and matrix assembly [287,288,289].

### 4.4. Proteomic Profiling of Fibrosis in the mdx-4cv Mouse Diaphragm

Because myofibrosis does not only directly impair tissue elasticity and the contractile functions of skeletal muscle fibers but also has a negative impact on the regenerative capacity of motor units following muscle degeneration and augments the susceptibility for muscular re-injury [10,15,16,17], it is a major contributing factor to muscle weakness in dystrophinopathy [290]. It was therefore of interest to determine biological-system-wide changes in X-linked muscular dystrophy using multi-omics approaches [291]. Proteomic profiling of dystrophic skeletal muscle has been carried out with a large number of proteomic approaches using both patient biopsy specimens and a variety of animal models of dystrophinopathy [292,293,294,295,296,297,298]. Proteome-wide changes include alterations in the abundance of muscle-associated proteins that are involved in the regulation of excitation–contraction coupling, ion homeostasis, cellular signaling, the contraction–relaxation cycle, cytoskeletal networks, the extracellular matrix, bioenergetic pathways, and the cellular stress response [299,300,301]. The extensive findings that have been generated by animal model research underline the critical importance of preclinical studies for increasing our mechanistic knowledge of muscle pathogenesis, establishing robust and reliable biomarker candidates, and developing novel treatment strategies for X-linked Duchenne muscular dystrophy [6,302,303,304].

Myofibrosis-associated changes in dystrophin-deficient muscle fibers are diagrammatically summarized in Figure 9, which highlights characteristic alterations in distinct protein species at the level of the sarcolemma, basal lamina, endomysium, perimysium, and epimysium. The figure is based on the proteomic profiling of the dystrophic *mdx-4cv* diaphragm muscle [55] and relates to the information given in Figure 4 above on the distribution of key extracellular matrix proteins. The almost-complete loss of dystrophin isoform Dp427-M was shown to affect all layers of the matrisome, including the basal lamina with the significant reduction in the laminin-binding protein alpha-dystroglycan. The loss of the trans-sarcolemmal linkage via the dystrophin/dystroglycan sub-complex is the key trigger of membrane micro-rupturing in dystrophinopathies. The elevated levels of matrisomal proteins at the level of the basal lamina are characterized by the increased abundance of collagen IV and nidogen-2. Changes in the endomysium are reflected by an increased concentration of collagen V and collagen VI. Additional extracellular matrix proteins are found throughout the endomysium, perimysium, and epimysium. Fibrosis-related increases were established for the matrisomal proteins fibronectin, vitronectin, and dermatopontin, as well as the non-structural matricellular protein periostin and the small leucine-rich proteoglycans asporin, biglycan, decorin, lumican, and mimecan/osteoglycin. Higher expression levels were also established for the extracellular-matrix-associated repair proteins annexin-2 and annexin-6 and the sarcolemmal integrin adhesion receptor subunits alpha-7 and beta-1.

Especially interesting is the drastic increase of periostin in the fibrotic *mdx-4cv* diaphragm, a matricellular protein that is involved in cellular signaling and the regulation of collagen fibrillogenesis [163,305]. In the extracellular matrix of skeletal muscles, periostin is only temporally expressed during regeneration and differentiation [306] but drastically increased during periods of fiber regeneration [307], including X-linked Duchenne muscular dystrophy [37,284,308]. This makes periostin a suitable fibrosis marker that can be used to judge the degree of myofibrotic changes in dystrophinopathy [284]. Changes in the annexin isoform ANAX2 [309,310,311] also have the potential to be highly useful for monitoring the level of membrane repair in fibrotic muscle fibers [312,313,314].

In summary, elevated levels of the below-listed major matrisomal marker proteins, which all play crucial roles in the extracellular matrix [73,315,316,317], were identified to occur during extended periods of myofibrosis. The detection and characterization of these protein species were carried out by various mass-spectrometry-based proteomic techniques in combination with biochemical, cell biological, and physiological analyses.

The following proteomic markers [37,55,284] could therefore be useful for the future establishment of an improved biomarker signature of dystrophinopathy-associated myofibrosis:(i)Basal-lamina-enriched extracellular matrix proteins:
Collagen IV (COL4A1, COL4A2, COL4A3);Collagen XV (COL15A1);Collagen XVIII (COL18A1);Nidogen-2/Osteonidogen (NID2).(ii)Endomysium-enriched extracellular matrix proteins:
Collagen V (COL5A1, COL5A2);Collagen VI (COL6A1, COL6A2, COL6A5, COL6A6).(iii)Matrisomal proteins of the endomysium, perimysium, and epimysium:
Fibronectin (FN1);Vitronectin (VTN);Dermatopontin (DPT).(iv)Small leucine-rich proteoglycans:
Asporin (ASPN);Biglycan (BGN);Decorin (DCN);Lumican (LUM);Mimecan/Osteoglycin (OGN).(v)Non-structural matricellular proteins:
Periostin (POSTN).(vi)Extracellular-matrix-associated repair proteins:
Annexin-2 (ANXA2);Annexin-6 (ANXA6).(vii)Adhesion receptors:
Integrin, alpha-7 (ITGA7);Integrin, beta-1 (ITGB1);Dystroglycan, alpha/beta (DAG1).(viii)Myotendinous-junction-enriched proteins:
Collagen XII (COL12A1).(ix)Tendon-enriched proteins:
Collagen I (COL1A1, COL1A2).

## 5. Conclusions

The suitability of the severely dystrophic and fibrotic *mdx-4cv* diaphragm as a pathophysiological surrogate to study the progressive nature of dystrophinopathies has been outlined in this article. Histological, cell biological, and mass-spectrometry-based proteomic studies of the aged *mdx-4cv* mouse diaphragm have confirmed the presence of extensive reactive myofibrosis in this genetic mouse model of Duchenne muscular dystrophy. In conjunction with progressive myonecrosis, central myonuleation, and chronic inflammation, the fibrotic changes in the senescent *mdx-4cv* diaphragm make it an excellent model to study X-linked muscular dystrophy. The *mdx-4cv* mouse model was shown to be suitable for high-throughput surveys of complex changes in the expression levels of collagens, proteoglycans, matricellular proteins, regulatory enzymes, and adhesion receptors of the extracellular matrix. Following independent verification of their sensitivity, specificity, and robustness, proteomic biomarkers can now be used to improve differential diagnostics, prognostics, and therapeutic monitoring of X-linked Duchenne muscular dystrophy.

## Figures and Tables

**Figure 1 biomolecules-13-01108-f001:**
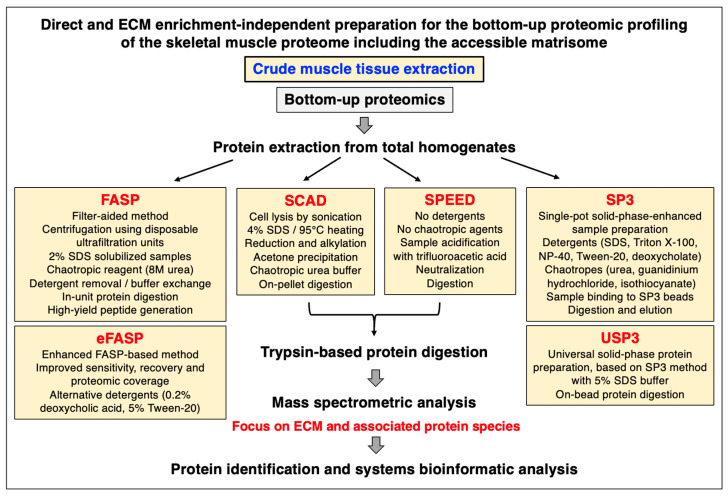
Overview of frequently used extracellular-matrix-enrichment-independent approaches for the bottom-up proteomic analysis of skeletal muscle specimens. Abbreviations used: ECM, extracellular matrix; FASP, filter-aided sample preparation; SCAD, surfactant-and-chaotropic-agent-assisted sequential extraction/on-pellet digestion; SDS, sodium dodecyl sulfate; SP3, single-pot solid-phase-enhanced sample preparation; SPEED, detergent-free sample preparation by easy extraction and digestion; USP3, universal solid-phase protein preparation.

**Figure 2 biomolecules-13-01108-f002:**
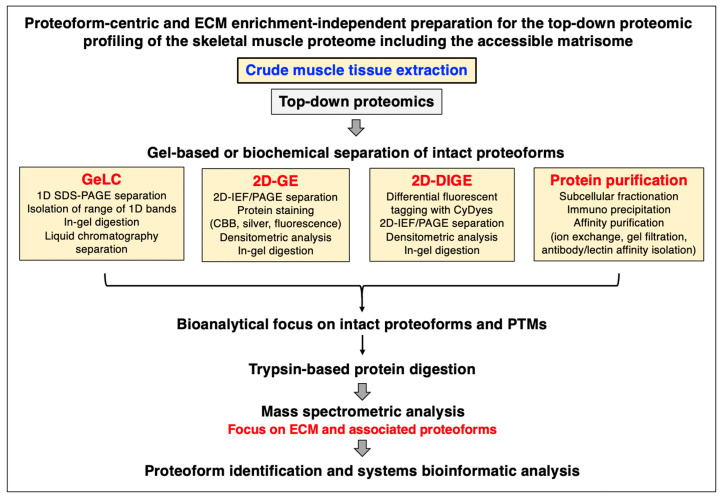
Outline of extracellular-matrix-enrichment-independent preparations for the proteoform-centric and top-down proteomic analysis of skeletal muscle specimens. Abbreviations used: 1D, one-dimensional; 2D, two-dimensional; DIGE, difference gel electrophoresis; ECM, extracellular matrix; GE, gel electrophoresis; GeLC, gel electrophoresis liquid chromatography; IEF, isoelectric focusing; PAGE, polyacrylamide gel electrophoresis; PTMs, post-translational modifications; SDS, sodium dodecyl sulfate.

**Figure 3 biomolecules-13-01108-f003:**
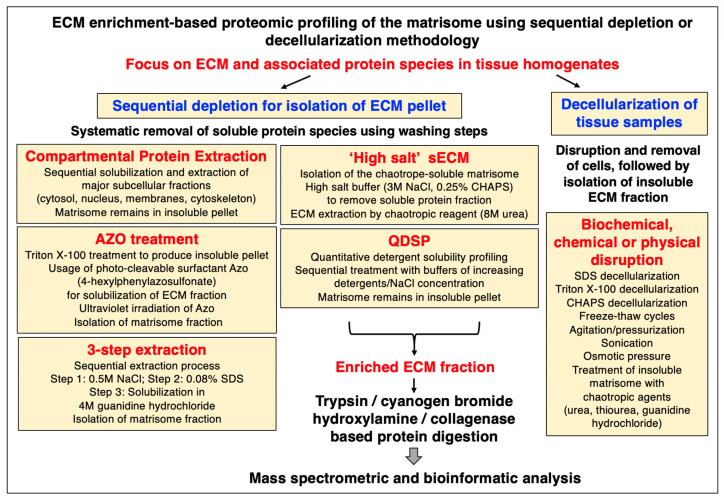
Outline of extracellular-matrix-enrichment-based approaches for the proteomic profiling of the matrisome. Abbreviations used: Azo, 4-hexylphenylazosulfonate; CHAPS, 3-[(3-cholamidopropyl) dimethylammonio]-1-propanesulfonate; ECM, extracellular matrix; QDSP, quantitative detergent solubility profiling; SDS, sodium dodecyl sulfate; sECM, soluble extracellular matrix.

**Figure 4 biomolecules-13-01108-f004:**
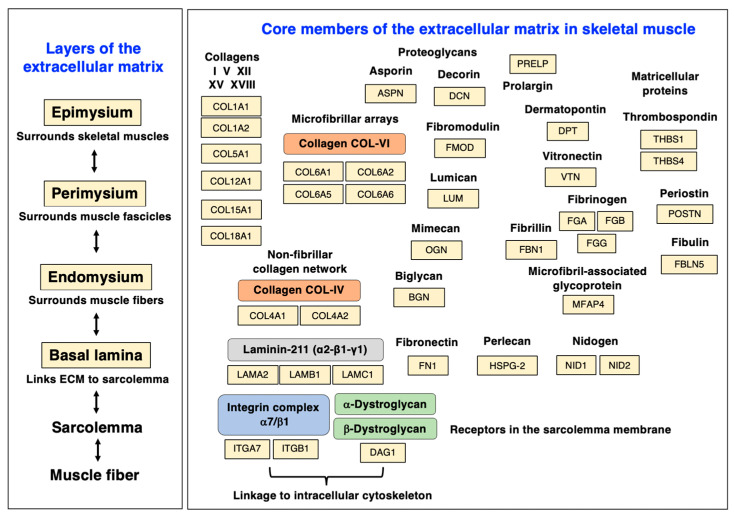
Diagram of the various layers of the extracellular matrix and distribution of major protein components of the matrisome in skeletal muscle tissues. Abbreviations used: ECM, extracellular matrix.

**Figure 5 biomolecules-13-01108-f005:**
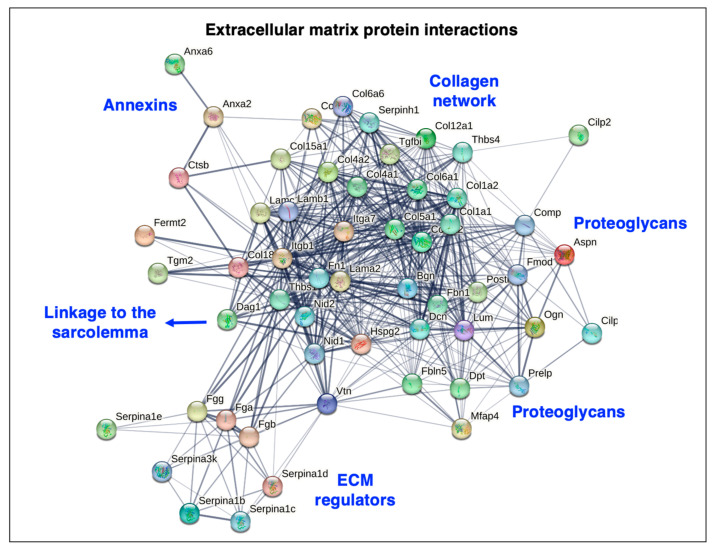
Bioinformatic STRING analysis of potential protein–protein interactions in the extracellular matrix of skeletal muscle. Matrisomal proteins that were identified by mass spectrometry, as listed in above Table 1, were analyzed by the method of Szklarczyk et al. [178]. The search list consisted of the matrisomal proteins that are encoded by the genes Lama2, Lamb1, Lamc1, Col4a1, Col4a2, Col15a1, Col18a1, Hspg2, Nid1, Nid2, Col5a1, Col6a1, Col6a2, Col6a5, Col6a6, Prelp, Dpt, Vtn, Fn1, Fga, Fgb, Fgg, Mfap4, Fbn1, Aspn, Bgn, Dcn, Fmod, Lum, Ogn, Postn, Thbs1, Thbs4, Fbln5, Itga7, Itgb1, Dag1, Col12a1, Fermt2, Col1a1, Col1a2, Comp, Cilp, Cilp2, Tgfbi, Tgm2, Anxa2, Anxa6, Serpina1d, Serpina1b, Serpina1c, Serpina1e, Serpinh1, Serpina3k, and Ctsb. Species was set to ‘*Mus musculus*’ with the following parameters: (i) full STRING network analysis where the edges indicate both functional and physical protein associations; (ii) full set of active interaction sources consisting of text mining, experiments, databases, co-expression, neighborhood, fusion, and co-occurrence; (iii) network edges using confidence whereby line thickness indicates the strength of data support; and (iv) network display mode with interactive scalable vector graphic. Abbreviations used: ECM, extracellular matrix.

**Figure 6 biomolecules-13-01108-f006:**
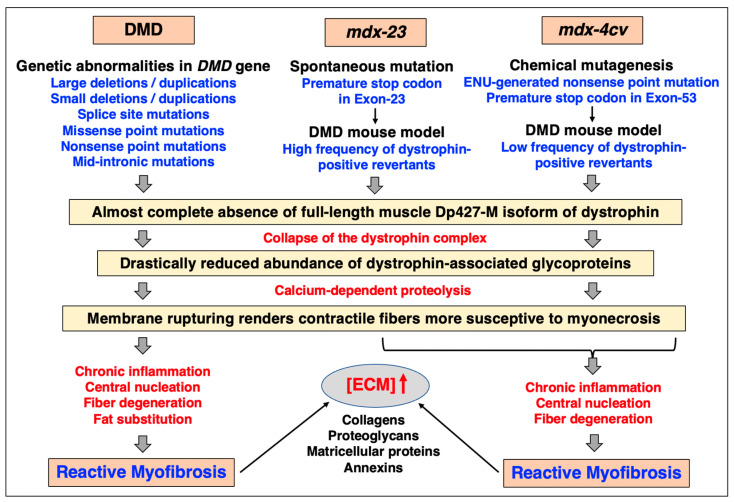
Comparative overview of the main features of Duchenne muscular dystrophy versus the *mdx-23* and the *mdx-4cv* models of dystrophinopathy that are due to a spontaneous mutation versus being generated by chemical mutagenesis, respectively. Analogous to Duchenne patients, the skeletal musculature of both the *mdx-23* and the *mdx-4cv* mouse harbor mutations in the *Dmd* gene, and exhibit progressive myonecrosis and chronic inflammation in combination with reactive myofibrosis, which is especially prominent in the aged diaphragm muscle. In contrast to the spontaneous *mdx-23* mouse, the *mdx-4cv* skeletal musculature is characterized by a low frequency of dystrophin-positive revertant fibers. Abbreviations used: DMD, Duchenne muscular dystrophy; ECM, extracellular matrix; ENU, N-ethyl-N-nitrosourea.

**Figure 7 biomolecules-13-01108-f007:**
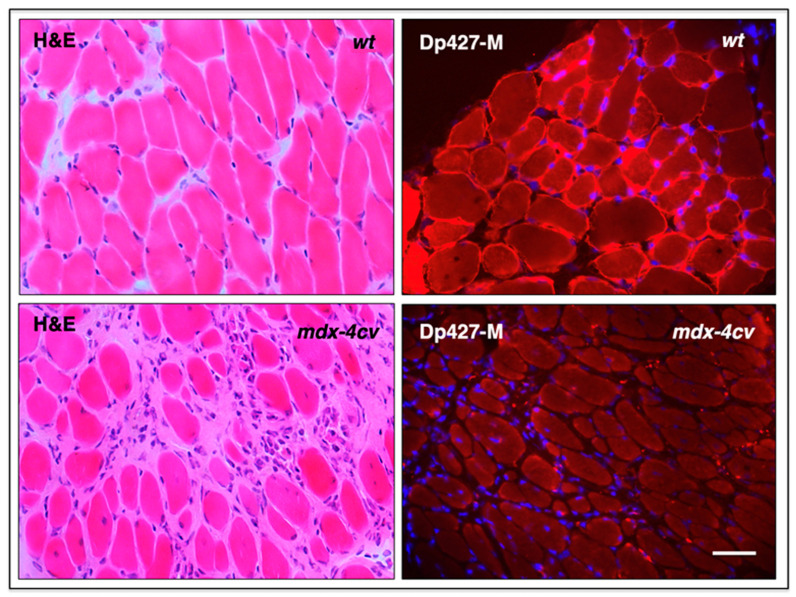
Histological and immunofluorescence microscopical analysis of dystrophic *mdx-4cv* mouse diaphragm muscle. Shown are transverse sections of 12-month-old diaphragm muscles from wild-type (*wt*) versus age-matched dystrophic *mdx-4cv* mice. Cryosections were stained with haematoxylin and eosin (H&E), as well as immuno-labelled with a monoclonal antibody to full-length dystrophin isoform Dp427-M. Hoechst-33342 staining was used to visualize myo-nuclei. Tissue preparation and immunofluorescence microscopy were carried out by optimized methods [146]. Bar equals 40 μm.

**Figure 8 biomolecules-13-01108-f008:**
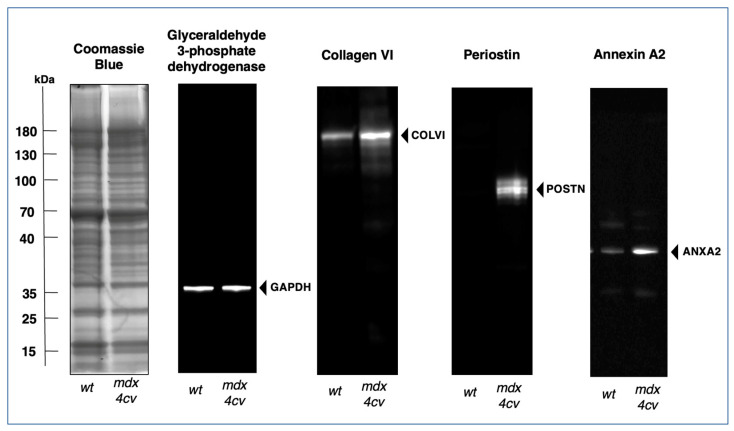
Comparative immunoblot analysis of the dystrophic *mdx-4cv* mouse diaphragm muscle. Shown is the immunoblotting of the glycolytic enzyme glyceraldehyde-3-phopshate dehydrogenase (GAPDH), used here as a house-keeping protein and loading control, alpha chain-containing collagen VI (COL-VI) of the endomysium as an abundant marker of the extracellular matrix, the matricellular protein periostin (POSTN), and the membrane repair protein annexin A2 (ANXA2). Gel electrophoretic separation was carried out with total protein extracts from aged wild-type (*wt*) versus aged and dystrophic *mdx-4cv* diaphragm. Lanes 1 and 2 contain protein extracts from 15-month-old *wt* and age-matched *mdx-4cv* muscle, respectively. On the left is shown a Coomassie Brilliant Blue-stained protein gel. The other images are identical immunoblots generated by electrophoretic transfer and then labelled with monoclonal antibodies to GAPDH, COL-VI, POSTN, and ANXA2, as previously described in detail [55]. The position of immuno-labelled protein bands is indicated by arrowheads. Molecular weight standards are marked on the left.

**Figure 9 biomolecules-13-01108-f009:**
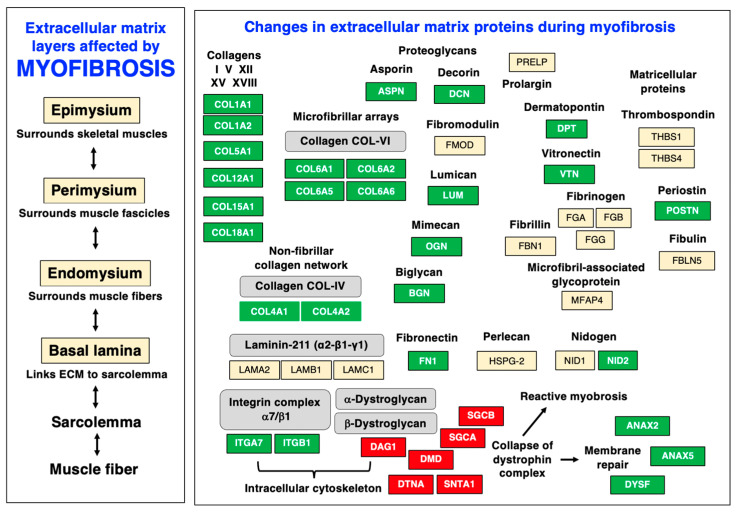
Diagram of myofibrosis-related changes within the various layers of the extracellular matrix and associated structures in dystrophin-deficient skeletal muscle. Increased versus decreased protein isoforms are shown with a green versus red background, respectively. The collapse of the dystrophin–glycoprotein complex results in compensatory mechanisms, such as the activation of membrane repair mechanisms and reactive myofibrosis, which is characterized by drastic increases in various collagens, matricellular proteins, and proteoglycans.

**Table 1 biomolecules-13-01108-t001:** Mass-spectrometry-based proteomic identification of key components of the various layers of the matrisome from diaphragm muscle *.

Accession Number	Extracellular Matrix Protein	Gene Name	Coverage (%)	Peptides	kDa
(i) **Basal-lamina-enriched extracellular matrix proteins**					
Q60675	Laminin-211, subunit alpha-2	Lama2	21.8	47	343.6
P02469	Laminin-211, subunit beta-1	Lamb1	14.9	16	197.0
P02468	Laminin-211, subunit gamma-1	Lamc1	30.2	31	172.2
P02463	Collagen IV, alpha-1	Col4a1	4.4	5	160.6
P08122	Collagen IV, alpha-2	Col4a2	2.5	3	167.2
O35206	Collagen XV, alpha-1	Col15a1	10.0	9	140.4
P39061-2	Collagen XVIII, alpha-1	Col18a1	2.8	3	134.1
Q05793	Perlecan (HSPG-2)	Hspg2	20.7	43	398.0
P10493	Nidogen-1 (entactin)	Nid1	30.1	18	136.5
O88322	Nidogen-2 (osteonidogen)	Nid2	5.9	6	153.8
(ii) **Endomysium-enriched extracellular matrix proteins**					
O88207	Collagen V, alpha-1	Col5a1	1.6	2	183.6
Q04857	Collagen VI, alpha-1	Col6a1	29.2	20	108.4
Q02788	Collagen VI, alpha-2	Col6a2	21.2	14	110.3
A6H584	Collagen VI, alpha-5	Col6a5	9.2	17	289.4
Q8C6K9	Collagen VI, alpha-6	Col6a6	12.5	16	246.2
(iii) **Extracellular matrix proteins throughout the endomysium, perimysium, and epimysium**					
Q9JK53	Prolargin	Prelp	25.7	7	43.3
Q9QZZ6	Dermatopontin (tyrosine-rich acidic matrix protein)	Dpt	19.4	2	24.0
P29788	Vitronectin	Vtn	6.7	2	54.8
P11276	Fibronectin	Fn1	15.8	22	272.4
E9PV24	Fibrinogen, alpha	Fga	24.2	12	87.4
Q8K0E8	Fibrinogen, beta	Fgb	63.6	24	54.7
Q8VCM7	Fibrinogen, gamma	Fgg	64.5	18	49.4
Q9D1H9	Microfibril-associated glycoprotein MFAP4	Mfap4	17.1	3	28.9
Q61554	Fibrillin-1	Fbn1	21.6	39	312.1
(iv) **Major muscle-associated small leucine-rich proteoglycans (SLRP type)**					
Q99MQ4	Asporin	Aspn	24.1	7	42.5
P28653	Biglycan	Bgn	26.6	7	41.6
P28654	Decorin	Dcn	34.5	11	39.8
P50608	Fibromodulin	Fmod	26.1	6	43.0
P51885	Lumican	Lum	27.2	6	38.2
Q62000	Mimecan (Osteoglycin)	Ogn	32.2	7	34.0
(v) **Matricellular proteins**					
Q62009-5	Periostin	Postn	9.6	4	87.0
P35441	Thrombospondin-1	Thbs1	15.4	12	129.6
Q9Z1T2	Thrombospondin-4	Thbs4	23.7	2	106.3
Q9WVH9	Fibulin-5	Fbln5	8.3	3	50.2
(vi) **Major sarcolemmal adhesion proteins/linkers to the extracellular matrix**					
Q61738-4	Integrin, alpha-7	Itga7	8.5	6	122.1
P09055	Integrin, beta-1	Itgb1	19.3	11	88.2
Q62165	Dystroglycan, alpha/beta	Dag1	4.1	3	96.8
(vii) **Myotendinous-junction-enriched proteins**					
Q60847-2	Collagen XII, alpha-1	Col12a1	18.5	36	333.5
Q8CIB5	FERM-domain containing kindlin 2 (FERMT-2)	Fermt2	6.5	3	77.8
(viii) **Tendon-enriched proteins**					
P11087	Collagen I, alpha-1	Col1a1	5.3	5	137.9
Q01149	Collagen I, alpha-2	Col1a2	5.0	5	129.5
(ix) **Cartilage-enriched proteins**					
Q9R0G6	Cartilage oligomeric matrix protein	Comp	3.7	2	82.3
Q66K08	Cartilage intermediate layer protein 1	Cilp	3.9	3	132.2
D3Z7H8	Cartilage intermediate layer protein 2	Cilp2	3.6	3	125.9
(x) **Extracellular matrix proteins involved in cellular signaling**					
P82198	Transforming growth factor-beta-induced protein ig-h3	Tgfbi	28.0	12	74.5
P21981	Protein-glutamine gamma-glutamyltransferase 2	Tgm2	36.0	15	77.0
(xi) **Extracellular-matrix-associated proteins belonging to the annexin family**					
P07356	Annexin A2	Anxa2	56.3	16	38.7
P14824	Annexin A6	Anxa6	51.0	23	75.8
(xii) **Extracellular matrix regulators**					
Q00897	Alpha-1-antitrypsin 1-4 (A1AT4)	Serpina1d	44.6	11	46.0
P22599	Alpha-1-antitrypsin 1-2 (A1AT2)	Serpina1b	38.0	9	45.2
Q00896	Alpha-1-antitrypsin 1-3 (A1AT3)	Serpina1c	35.9	9	45.8
Q00898	Alpha-1-antitrypsin 1-5 (A1AT5)	Serpina1e	26.2	8	45.9
P19324	Serpin H1 (SERPH)	Serpinh1	9.6	3	46.5
P07759	Serine protease inhibitor A3K	Serpina3k	26.8	8	46.8
P10605	Cathepsin B	Ctsb	22.4	5	37.3

* The table lists major protein hits from the mass spectrometric analysis of wild-type mouse diaphragm specimens, analyzed by standardized bottom-up proteomic methodology [96,146] with the help of the detergent-based and filter-aided sample preparation (FASP) method [86,87]. Proteins were selected on high-confidence amino acid sequence coverage with a peptide number of 2 and above for sequence recognition. Listed are the accession number of detected proteins, the names of identified extracellular matrix proteins, the corresponding gene name, the percentage sequence coverage, the number of identified peptides, and the predicted molecular mass in kDa.

**Table 2 biomolecules-13-01108-t002:** Comparison of major genetic mouse models of Duchenne muscular dystrophy *.

Mouse Model	Genetic Alteration	Affected Dystrophin Isoforms	Dystrophic Phenotype
*mdx (mdx-23)*	Exon-23 of *Dmd* gene; spontaneous premature stop codon	Dp427	Moderate, but severely affected diaphragm; degeneration–regeneration cycles; considerable number of revertant muscle fibers; mildly affected heart muscle
*mdx/Dtna*	Exon-23 of *Dmd* gene; plus dKO of *Dtna* gene	Dp427	Severely affected general musculature; moderately affected heart muscle
*mdx/Cmah*	Exon-23 of *Dmd* gene; plus Exon-6 deletion of *Cmah* gene	Dp427	Severely affected general musculature; moderately affected diaphragm and heart
*mdx/Utr*	Exon-23 of *Dmd* gene; plus targeted disruption of *Utrn* gene	Dp427; plus lacking autosomal dystrophin homologue utrophin (Up395)	Severely affected general musculature including diaphragm; moderately affected heart muscle
*mdx/α7*	Exon-23 of *Dmd* gene; plus dKO of *Itga7* gene	Dp427	Severely affected general musculature including diaphragm; mildly affected heart muscle
*mdx/Myod1*	Exon-23 of *Dmd* gene; plus dKO of *Myod1* gene	Dp427	Severely affected general musculature including diaphragm, and heart muscle
*mdx-2cv*	Intron-42 of *Dmd* gene; ENU-mutagenesis-induced shift in reading frame	Dp427, Dp260	Moderate with large variation in muscle fiber size; severely affected diaphragm
*mdx-3cv*	Intron-65 of *Dmd* gene; ENU-mutagenesis-induced shift in reading frame	Dp427, Dp260, Dp140, Dp116, Dp71	Moderate, but severely affected diaphragm; no revertant fibers
*mdx-4cv*	Exon-53 of *Dmd* gene; ENU-mutagenesis-induced premature stop codon	Dp427, Dp260, Dp140	Moderate with large variation in muscle fiber size; severely affected diaphragm; fewer revertant fibers than *mdx-23* model
*NSG-mdx-4cv*	Exon-54 of *Dmd* gene; immunodeficient muscular dystrophy model	Dp427, Dp260, Dp140	Moderate, but severely affected diaphragm; fewer revertant fibers than *mdx-23* model
*mdx-4cv/mTR-G2*	Exon-53 of *Dmd* gene; KO of mTR; model with humanized telomere lengths	Dp427, Dp260, Dp140	Severely affected general musculature including diaphragm, and heart muscle
*mdx-5cv*	Exon-10 of *Dmd* gene; ENU-mutagenesis-induced frame shift deletion	Dp427	Moderate, but severely affected diaphragm; fewer revertant fibers than *mdx-23* model
*mdx-52*	Exon-52 of *Dmd* gene; targeted disruption induced point mutation	Dp427, Dp260, Dp140	Moderate, but severely affected diaphragm; fewer revertant fibers than *mdx-23* model
*mdx-βgeo*	Intron-63 of *Dmd* gene; insertion of β-geo gene trap cassette	Dp427, Dp260, Dp140, Dp116, Dp71	Moderately affected general musculature
*Dmd-null*	*Dmd* gene deletion; Cre-loxP system of entire *Dmd* gene	Dp427, Dp260, Dp140, Dp116, Dp71	Severely affected general musculature
*hDMD/mdx-45*	Spontaneous Exon-23 mutation of *Dmd* gene; plus CRISPR/Cas-mediated Exon-45 deletion in *hDMD* gene	Dp427, Dp260, Dp140 (murine and human)	Moderate, but severely affected diaphragm; revertant muscle fibers
*hDMD/mdx-52*	Spontaneous Exon-23 mutation of *Dmd* gene; plus TALEN-based partial deletion of Exon-52 in *hDMD* gene	Dp427, Dp260, Dp140 (murine and human)	Moderate, but severely affected diaphragm; revertant muscle fibers

* Abbreviations used: Cre-loxP, causes recombination (site-specific recombinase enzyme of bacteriophage P1)-locus of X-over P1; CRISPR/Cas, clustered regularly interspaced short palindromic repeats/CRISPR-associated protein; Dp, dystrophin protein; ENU, N-ethyl-N-nitrosourea; *hDMD*, human Duchenne muscular dystrophy gene; mTR, telomerase RNA; TALEN, transcription activator-like effector nuclease.

**Table 3 biomolecules-13-01108-t003:** List of major studies on the generation and characterization of the *mdx-4cv* mouse model of Duchenne muscular dystrophy.

Experimental Focus	Bioanalytical Approach	Major Findings	References
(i) **Generation, genotyping, and phenotyping of the *mdx-4cv* model**			
Generation of *mdx-4cv* model	Chemical mutagenesis with N-ethyl-N-nitrosourea (ENU)	Induction of a C-to-T transition at position 7916 in exon-53 of the *Dmd* gene leading to premature translation termination	[41]
Genotyping of *mdx-4cv* model	DNA sequencing, polymerase chain reaction analysis	Protein-truncating nonsense mutation that introduces premature stop codon in exon-53 of *Dmd* gene	[42,43,44,237]
Characterization of *mdx-4cv* model	Cell biological and biochemical analyses	Less revertant fibers compared to spontaneous *mdx-23* mouse; reduced level of dystrophin–glycoprotein complex	[231]
(ii) **Evaluation of experimental therapies using the *mdx-4cv* model**			
Exon-skipping therapy	Antisense molecule-based skipping of defective exon	Removal of exon-53 by-passed the protein-truncating mutation and restored the synthesis of semi-functional Dp427-M protein	[238]
Micro-dystrophin gene therapy	Virus- or nano-carrier mediated delivery of micro-dystrophin gene	Increased numbers of micro-dystrophin positive fibers using a variety of delivery mechanisms	[239,240,241,242,243]
Cell-based therapy	Cell transplantation to introduce mini-dystrophin protein in dystrophic fibers	High levels of mini-dystrophin expression, but no physiological improvements	[244,245,246]
Genome editing	CRISPR/Cas9 (clustered regularly interspaced short palindromic repeats)/CRISPR-associated protein 9)	Increased dystrophin expression in CRISPR/Cas9-treated muscles and increased force generation.	[247,248]
(iii) **Proteomic profiling of skeletal muscle tissue specimens from the *mdx-4cv* model**			
Skeletal muscle, hindlimb	Liquid chromatography– tandem mass spectrometry (LC-MS/MS)	Increased collagen VI, fibronectin, fibrinogen, asporin, annexin-2; reduced dystrophin complex, carbonic anhydrase CA3, parvalbumin, myozenin-2	[229]
Diaphragm muscle	LC-MS/MS	Increased extracellular matrix proteins (collagens, annexins, proteoglycans) and molecular chaperones; decreased dystrophin complex, parvalbumin, carbonic anhydrase CA3, excitation–contraction coupling proteins	[55,146]
Extraocular muscle	LC-MS/MS	Mild phenotype lacking drastic changes in protein abundance	[252]
Subcellular skeletal muscle fractions	Subcellular fractionation, affinity purification, LC-MS/MS	Increased membrane repair proteins (myoferlin, dysferlin, annexins) and extracellular matrix proteins (collagens); decrease in dystrophin complex	[230,251]
Skeletal muscle protein fractions	Chemical crosslinking analysis	Altered patterns of protein interactions in dystrophin-deficient fibers	[249,250]
(iv) **Proteomic profiling of non-skeletal muscle tissue specimens from the *mdx-4cv* model**			
Heart	LC-MS/MS	Decreased dystrophin complex; identified changes in laminin, periostin, asporin, and lumican, heat shock proteins, mitochondrial and glycolytic enzymes	[253]
Liver	LC-MS/MS	Elevated levels of fatty-acid-binding protein FABP5; changes in proteins involved in fatty acid, carbohydrate, and amino acid metabolism	[254]
Kidney	LC-MS/MS	Elevated levels of fatty-acid-binding protein FABP1; complex changes in metabolic and bioenergetic enzymes	[255,256]
Stomach/pancreas interface	LC-MS/MS	Identification of dystrophin complex in normal stomach muscles; reduced dystrophin complex in *mdx-4cv* stomach/pancreas-interface	[258]
Spleen	LC-MS/MS	Identification of short dystrophin isoform in spleen; altered proteins involved in metabolism, signaling, and cellular architecture; crosstalk between lymphoid system and muscle	[96,257]
Brain	LC-MS/MS	Increased levels of gliosis marker GFAP (glial fibrillary acidic protein); altered abundance of a variety of neuronal proteins	[259]
(v) **Proteomic profiling of biofluids from the *mdx-4cv* model**			
Serum	LC-MS/MS	Increased levels of various muscle damage markers in serum; high levels of the inflammation-induced plasma marker haptoglobin	[260,261]
Saliva	LC-MS/MS	Increased levels of kallikrein Kkl-1 and the Klk1-related peptidases Klk1-b1, Klk1-b5 and Klk-b22	[261,263]
Urine	LC-MS/MS	Increased levels of various muscle damage markers in urine; high levels of titin fragments	[262]

## Data Availability

The mass spectrometric raw data from studies of the extracellular matrix from skeletal muscle shown in tables are available upon request.

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
