# Peer review of "Extracellular Matrix Proteomics: The *mdx-4cv* Mouse Diaphragm as a Surrogate for Studying Myofibrosis in Dystrophinopathy"

_biomolecules, 2023, doi:10.3390/biom13071108_

Round 1
Reviewer 1 Report
In this review, the authors have summarized various mass spectrometry-based strategies in identifying extracellular matrix proteins and discuss their relevance in the DMD mice model – ENU generated mdx-4cv. This is an interesting and comprehensive review of proteomics, although it is pretty long. The authors have firmly taken the view that mdx-4cv mice have the best phenotype concerning fibrosis in the aged mice and present research results conducted on this mice model to support this notion.
The manuscript provides a thorough background on fibrosis in mdx-4cv and DMD patients. The review falls short in several areas, most notably that there is no comparison between mdx-4cv to other DMD mouse models. The figures and tables could also be more informative. For these reasons, the manuscript is not suitable for publication in its present form.
Other comments:
- In line 88, the authors mention the abnormal processes within the epimysium, perimysium, and endomysium but do not develop their review in this direction. In Fig 4, they have only shown the distribution of proteins but did not discuss abnormal processes.
- In section 2, the authors cite some publications demonstrating SWATH-MS use, but none related to DMD.
- In section 3, the authors did not discuss the post-translational modification of extracellular matrix protein.
- Section 4.3 starts with a title about proteomic profiling but mainly discusses mice phenotype, and proteomics is in the last section.
- The H&E staining in Fig. 7 shows very few centralized nuclei. Is it typical for this particular type of DMD model?
Author Response
Thanks for reviewing our manuscript [biomolecules-2469095] entitled ‘Extracellular matrix proteomics: the mdx-4cv mouse diaphragm as a surrogate for studying myofibrosis in dystrophinopathy’. We appreciate the time taken by Reviewer 1 to critically examine our submission. We have carried out a detailed revision of our paper and have listed our responses to individual points made by Reviewer 1 in below point-by-point responses. A copy of the revised manuscript with highlighted changes has been uploaded to illustrate the changes made in the R1 version of this paper.
Reviewer 1, Comment 1: ‘In this review, the authors have summarized various mass spectrometry-based strategies in identifying extracellular matrix proteins and discuss their relevance in the DMD mice model – ENU generated mdx-4cv. This is an interesting and comprehensive review of proteomics, although it is pretty long. The authors have firmly taken the view that mdx-4cv mice have the best phenotype concerning fibrosis in the aged mice and present research results conducted on this mice model to support this notion. The manuscript provides a thorough background on fibrosis in mdx-4cv and DMD patients. The review falls short in several areas, most notably that there is no comparison between mdx-4cv to other DMD mouse models. The figures and tables could also be more informative. For these reasons, the manuscript is not suitable for publication in its present form’.
Response: To address the points in this comment, we have revised our manuscript accordingly and now compare the mdx-4cv mouse to other mouse models of DMD. Additional information has been supplied in relation to figures and tables, which also addresses points made by Reviewer 2.
The comparison of the mdx-4cv mouse to other genetic mouse models of dystrophinopathy includes additional text and a new table, Table 2, as well as new references that focus on mouse models of muscular dystrophy. The new text section and new Table 2 have been introduced in Section 4.2. on Pages 14-15 in the revised manuscript summarizing the different types of mouse models of DMD. The other tables and references have been renumbered accordingly.
This new table lists information on the names of DMD mouse models, their specific genetic alterations, the affected dystrophin isoforms and the dystrophic phenotype. The table compares the original mdx-23 model to mdx/Dtna, mdx/Cmah, mdx/Utr, mdx/alpha7, mdx/Myod1, mdx-2cv, mdx-3cv, mdx-4cv, mdx-4cv/mTR-G2, mdx-5cv, mdx-52, mdx-beta-geo, Dmd-null, hDMD/mdx-45 and hDMD/mdx-52 models.
Revised text on Pages 15-19 (including new Table 2; please see revised R1 version):
‘4.2. Characterization of the mdx-4cv mouse within the context of other dystrophic models
Both spontaneous and bioengineered mouse models that lack the dystrophin protein isoform Dp427-M have been instrumental for the detailed elucidation of the molecular and cellular pathogenesis of Duchenne muscular, as well as the evaluation of experimental treatment strategies [210-212]. A comparative listing of major genetic mouse models of dystrophinopathy is provided in Table 2. This includes the original mdx-23 mouse as compared to a variety of modified models including mdx/Dtna, mdx/Cmah, mdx/Utr, mdx/alpha7, mdx/Myod1, mdx-2cv, mdx-3cv, mdx-4cv, mdx-4cv/mTR-G2, mdx-5cv, mdx-52, mdx-beta-geo, Dmd-null, hDMD/mdx-45 and hDMD/mdx-52. The table lists information on the genetic abnormalities that characterize the various mouse models and the observed severity of the dystrophic phenotype in individual models of Duchenne muscular dystrophy. Importantly, Table 2 gives an overview of the effect of mutations on the expression of distinct dystrophin isoforms ranging from Dp71 to Dp427. Depending on the genetic abnormality within the Dmd gene, impaired expression patterns might only cause the loss of the full-length dystrophins Dp427-M, Dp427-B and Dp427-P, or also additional effects on the tissue-specific density of the shorter dystrophins Dp260-R, Dp140-B/K, Dp116-S, Dp71 and/or Dp45. Detailed comparisons of dystrophic mouse models have been carried out in recent reviews [213-215].
Compared to the onset of dystrophic changes in the skeletal musculature at approximately 1-month of age in the mdx-23 mouse and a maximum lifespan of approximately 2 years, the modified mdx-type mice differ considerably in the onset of histopathological changes, ranging from 2-8 weeks, and their longevity, ranging in lifespan from 1-23 months [216]. Both the spontaneously mutated mdx-23 mouse and the chemically mutated mdx-4cv mouse have comparable survival rates and similar onsets of moderate hindlimb degeneration, as judged by histopathology, at 3-4 weeks of age in association with physiological dysfunction such as reduced muscle force. The mdx-23 mouse is characterized by extensive cycles of degeneration and regeneration from 3-8 weeks, illustrated by high levels of central myonucleation, and some stabilization with age [38-40]. The mdx-4cv musculature shows large variation in skeletal muscle fiber sizes. In both model systems, the dystrophin-deficient diaphragm is severely affected starting at 3-4 weeks of age and shows high levels of reactive myofibrosis during aging [213-215]. In analogy to the skeletal musculature, the reduced presence of the cardiac dystro-phin-glycoprotein complex causes sarcolemmal disintegration, myonecrosis, fatty tissue replacement, fibrotic scarring and interstitial inflammation in the mdx-type heart [217]. The onset of cardiomyopathy is at approximately 6-9 months of age and associated with considerable proteome-wide changes during aging [218].
The range of modified mdx-type mouse models of dystrophinopathy with usually more severe phenotypes includes double mutants that affect besides dystrophin levels also the expression of proteins such as the dystrophin-associated component dystrobrevin, the autosomal dystrophin homologue utrophin, myogenic differentiation protein MOD1 and alpha7-integrin [213-216]. In addition to the most frequently used murine models with their rapid disease progression, large animal models of Duchenne muscular dystrophy exist in the form of canine and porcine mutants [219-221]. The Golden Retriever muscular dystrophy (GRMD) model exhibits a splice site mutation in intron-6, which causes skipping of exon-7 and a resulting out-of-frame DMD transcript in exon-8 [220]. The complexity and progression of the clinical syndrome is severe in the GRMD model including cardiomyopathic complications making dystrophic dogs good model systems for translational studies [222,223]. The porcine model of dystrophinopathy (DMD pig) is characterized by a deletion in Exon-52 of the DMD gene and exhibits progressive cardiomyopathy [224]. In conjunction with mouse models, larger animals can be utilized to study pathophysiological mechanisms, validate the suitability and robustness of novel biomarker candidates, and be helpful during the preclinical testing phase of new therapeutic approaches such as gene editing [219,221]. Large disease model animals are highly useful for facilitating the efficient translation of novel diagnostic methods to the clinical setting, such as imaging technology for the monitoring of myofibrosis [225].
Table 2: Comparison of major genetic mouse models of Duchenne muscular dystrophy*.
Skeletal muscle fibrosis is seen in different mdx-type muscles to varying degrees [46-48,226], with the diaphragm being the most structurally affected and fibrotic muscle tissue [49-52,227,228]. However, one of the most widely employed dystrophic animal models, the mdx-23 mouse [38], exhibits a relatively high frequency of revertant fibers [39] and a relatively mild dystrophic phenotype [40]. In contrast, the chemically mutated mdx-4cv mouse which harbors a nonsense mutation in exon-53 [41-43] has a drastically reduced number of dystrophin-positive revertants [45]. Figure 6 provides a comparison of the main genetic, biochemical, physiological and histological features of muscular abnormalities in Duchenne muscular dystrophy versus the widely used mdx-23 and mdx-4cv mouse models of dystrophinopathy’.
Since Reviewer 2 although raised issues with certain figures and tables, additional information and revisions were carried out to improve the presentation of our manuscript. As also outlined in detail to individual points raised by Reviewer 2, this included:
- Better discussion of the proteins listed in Table 1 in the section describing the various ECM protein families.
- Figure 5 was revised to improve the ECM network presentation and additional information was provided on the STRING analysis in the figure legend.
- A new Table 2 was integrated in the revised text on the comparison of major genetic mouse models of Duchenne muscular dystrophy. A discussion of large animal models (DMD pig and GRMD dog) has also been added to the revised manuscript.
- The sub-headings within Table 3 (old Table 2) were revised.
- A new Figure 9 was added to the section on the proteomic profiling of fibrosis in the mdx-4cv mouse diaphragm. This figure summarizes the myofibrosis-related changes within the various layers of the extracellular matrix and associated structures in dystrophin-deficient skeletal muscle.
Reviewer 1, Comment 2: ‘Other comments: 1. In line 88, the authors mention the abnormal processes within the epimysium, perimysium, and endomysium but do not develop their review in this direction. In Fig 4, they have only shown the distribution of proteins but did not discuss abnormal processes’.
Response: To address this point, we have added a new summarizing Figure 9 and additional discussion of the affected matrisomal proteins and their distribution within the various layers of the ECM in revised and re-numbered Section 4.4. on the proteomic profiling of fibrosis in the mdx-4cv mouse diaphragm. Figure 9 relates to introductory Figure 4 and highlights changes in specific protein species and their position within the sarcolemma, basal lamina, epimysium, perimysium, and endomysium. Increased versus decreased protein isoforms are shown with a green versus red background, respectively.
New text on Pages 24-25: ‘Myofibrosis-associated changes in dystrophin-deficient muscle fibers are diagrammatically summarized in Figure 9, which highlights characteristic alterations in distinct protein species at the level of the sarcolemma, basal lamina, endomysium, perimysium and epimysium. The figure is based on the proteomic profiling of the dystrophic mdx-4cv diaphragm muscle [55] and relates to the information given in above Figure 4 on the distribution of key extracellular matrix proteins. The almost complete loss of dystrophin isoform Dp427-M was shown to affect all layers of the matrisome, including the basal lamina with the significant reduction in the laminin-binding protein alpha-dystroglycan. The loss of the trans-sarcolemmal linkage via the dystrophin/dystroglycan sub-complex is the key trigger of membrane micro rupturing in dystrophinopathies. The elevated levels of matrisomal proteins at the level of the basal lamina are characterized by increased abundance of collagen-IV and nidogen-2. Changes in the endomysium are reflected by an increased concentration of collagen-V and collagen-VI. Additional extracellular matrix proteins are found throughout the endomysium, perimysium and epimysium. Fibrosis-related increases were established for the matrisomal proteins fibronectin, vitronectin and dermatopontin, as well as the non-structural matricellular protein periostin and the small leucine-rich proteoglycans asporin, biglycan, decorin, lumican and mimecan/osteoglycin. Higher expression levels were also established for the extracellular matrix-associated repair proteins annexin-2 and annexin-6 and the sarcolemmal integrin adhesion receptor subunits alpha-7 and beta-1’.
New Figure 9 (Page 25; please see revised R1 version). Diagram of myofibrosis-related changes within the various layers of the extracellular matrix and associated structures in dystrophin-deficient skeletal muscle. Increased versus decreased protein isoforms are shown with a green versus red background, respectively. The collapse of the dystrophin-glycoprotein complex results in compensatory mechanisms, such as the activation of membrane repair mechanisms and reactive myofibrosis, which is characterized by drastic increases in various collagens, matricellular proteins and proteoglycans.
Reviewer 1, Comment 3: ‘2. In section 2, the authors cite some publications demonstrating SWATH-MS use, but none related to DMD’.
Response: The revised text now contains a new reference and description of SWATH-MS analysis of dystroglycanopathy, a disorder related to abnormalities in the dystrophin-glycoprotein complex.
Revised Page 7: ‘ … the sequential window acquisition of all theoretical mass spectra mass spectrometry (SWATH-MS) method, which divides the mass range into small mass windows, has been successfully applied to studying extracellular matrix proteins [132]. Immunoprecipitation combined with SWATH-MS was successfully employed to investigate fukutin-related protein (FKRP)-associated dystroglycanopathy [133]. This methodology revealed that loss of FKRP function disrupts the localization and tethering of fibronectin to myosin-10, essential for fibronectin maturation and subsequent collagen binding. The evidence accumulated demonstrates the utility of combining SWATH-MS and enrichment methods to investigate distinct phenotypes associated with muscular disorders’.
New reference [133]: Wood, A.J.; Lin, C.H.; Li, M.; Nishtala, K.; Alaei, S.; Rossello, F.; Sonntag, C.; Hersey, L.; Miles, L.B.; Krisp, C.; et al. FKRP-dependent glycosylation of fibronectin regulates muscle pathology in muscular dystrophy. Nat. Commun. 2021, 12, 2951.
The remaining references were re-numbered accordingly.
Reviewer 1, Comment 4: ‘3. In section 3, the authors did not discuss the post-translational modification of extracellular matrix protein’.
Response: In response to this comment, we have introduced information on PTMs in ECM proteins in revised Section 3, as well as new references on this topic.
Revised text on Pages 7-8: ‘… maps of the matrisome. Importantly, PTMs profiles of extracellular matrix proteins can provide important biochemical characteristics for detecting and understanding the complex pathways associated with skeletal muscle physiology and pathophysiology. Mounting evidence exists showing how the existence of various PTMs are essential for matrisomal protein secretion and functionality within the extracellular milieu. The thrombospondin type 1 domain (TSR) is a common molecular feature present in the thrombospondin family, where O-fucosylation has a crucial stabilizing effect on the TSR structure [142]. Methods for detecting hydroxylation and glycosylation sites in collagen-IV have previously been documented [143]. Such methods are important steps for understanding the role of specific PTMs in health and disease, as hydroxylation is essential for the stability of collagen molecules, and associated fibril structures. Disulfide bonds, assembled through the action of protein disulfide isomerases (PDIs), are found in thrombospondins and fibronectin, playing important roles in the establishment of subunit stability [144]. Citrullination, in which the conversion of the amino acid arginine into the amino acid citrulline is established, is associated with several extracellular matrix proteins, including fibronectin, having important roles in protein clustering, focal adhesion stability and signaling [145]’.
New references on PTMs in ECM proteins:
[142] Schneider, M.; Al-Shareffi, E.; Haltiwanger, R.S. Biological functions of fucose in mammals. Glycobiology. 2017, 27, 601-618.
[143] Basak, T.; Vega-Montoto, L.; Zimmerman, L.J.; Tabb, D.L.; Hudson, B.G.; Vanacore, R.M. Comprehensive Characterization of Glycosylation and Hydroxylation of Basement Membrane Collagen IV by High-Resolution Mass Spectrometry. J. Proteome Res. 2016, 15, 245-258.
[144] Galligan, J.J.; Petersen, D.R. The human protein disulfide isomerase gene family. Hum. Genomics. 2012, 6, 6.
[145] Stefanelli, V.L.; Choudhury, S.; Hu, P.; Liu, Y.; Schwenzer, A.; Yeh, C.R.; Chambers, D.M.; von Beck, K.; Li, W.; Segura, T.; et al. Citrullination of fibronectin alters integrin clustering and focal adhesion stability promoting stromal cell invasion. Matrix Biol. 2019, 82, 86-104.
The remaining references were re-numbered accordingly.
Reviewer 1, Comment 5: ‘4. Section 4.3 starts with a title about proteomic profiling but mainly discusses mice phenotype, and proteomics is in the last section’.
Response: We agree and have altered the sub-section structure of Section 4 accordingly. Original Section 4.3. has been split into two sections, i.e. ‘4.3. Histological and biochemical characterization of fibrosis in the mdx-4cv mouse diaphragm’ and ‘4.4. Proteomic profiling of fibrosis in the mdx-4cv mouse diaphragm’ in the revised manuscript. The original first paragraph has been moved to new Section 4.4. References have been re-numbered accordingly. New Section 4.3. outlines now the cell biological and biochemical characterization of fibrosis in the mdx-4cv model showing histological images in Figure 7 and immunoblotting in Figure 8. New Section 4.4. focuses on proteomic findings and summarizes in new Figure 9 the main changes in the matrisome during reactive myofibrosis.
New Section 4.3: Histological and biochemical characterization of fibrosis in the mdx-4cv mouse diaphragm.
New Section 4.4: Proteomic profiling of fibrosis in the mdx-4cv mouse diaphragm.
Reviewer 1, Comment 6: ‘5. The H&E staining in Fig. 7 shows very few centralized nuclei. Is it typical for this particular type of DMD model?’.
Response: Due to the relative thin structure of transversely cut muscle tissue section, these preparations only capture a sub-population of fibers with central nucleation. Some of the central nuclei are relatively weakly stained as compared to peripherally located nuclei. However, the image presented in Figure 7 shows over 10 centrally located myonuclei in approximately 60 displayed individual muscle fibers. In our experience, this is typical for 12-month old diaphragm muscle specimens from the dystrophic mdx-4cv mouse. The age of the mice used for this analysis is now provided in the revised figure legend of Figure 7.
Revised figure legend of Figure 7: ‘Histological and immunofluorescence microscopical analysis of dystrophic mdx-4cv mouse diaphragm muscle. Shown are transverse sections of 12-month old diaphragm muscles from wild type (wt) versus age-matched dystrophic mdx-4cv mice. Cryosections were …’.
Reviewer 2 Report
In this review, the authors make a considerable effort to present the components of the skeletal muscle matrisome at different anatomical levels (epimysium, perimysium, endomysium, MTJ, tendon etc) based on their studies and existing databases and literature (for example the widely used Matrisome project from MIT). The review starts with a useful and detailed technical overview of the different proteomic-based approaches to extract and analyse the extracellular matrix proteins.
The role of the extracellular matrix in muscular dystrophies is paramount and so the information is very relevant.
However, the information presented in the text, figures and tables is sometimes disconnected and the scope of the MS and the criteria to select the information in the review are not clear and appear a little random.
For example, in section 3 the authors select a set of 25 proteins and list them in Table 1. According to the legend, these are the major proteins that they have identified in a previous study in the control mouse diaphragm. However, in the text right after the table, the authors describe the major families of extracellular matrix components without an apparent relationship with Table 1.
The order in which proteins are listed in Table 1 should be mentioned. Are they listed according to the number of peptides? Coverage?. Why were these proteins selected and not others?
I suggest focusing that section on describing the proteins in Table 1 if that is the scope of the review (diaphragm).
Similarly for Figure 4. Which criteria were used to select the proteins to be represented (what is the source)? What is the meaning of the spatial distribution of the proteins in the boxes? This refers to the composition of skeletal muscle in general and not just the diaphragm.
Regarding the STRING network in Figure 5. Which list of proteins was selected to construct the network? There are only 25 proteins in Table 1 and in figure5 there are more than 25 nodes (proteins). What settings were used to construct the network? STRING allows for different parameters to build the edges depending on the type of interaction and level of confidence and this should be stated. Also, other descriptors such as the enrichment index (number of interactions or edges, PPI enrichment, could be used). The figure is very condensed and small and it is difficult to visualize the different clusters pointed out in the text by the authors. Perhaps unsupervised clustering analysis could be performed.
In section 4 the authors focus on the relevance of the mdx-4cv mouse model in the study of DMD.
The purpose of listing experimental therapies in this mouse model is not clear and seems out of context. Given the aim of the review, I suggest focusing on the list of proteomic studies performed in this mouse model in skeletal muscle and biofluids. The data on other tissues do not seem to provide additional information.
In section 4.3 there are some parts where the authors change from describing the fibrosis in mdx-4cv to then talking in general about changes detected by western blot in other models and DMD and I find this confusing.
The criteria for the selection of proteins in Figure 8 and the list at the end of the results section are not clear. Are those the most differentially abundant proteins identified between wt and mdx-4cv diaphragms?
Minor points
Section 4.3 COL-VI isoform: Collagen VI is a collagen type made of alpha-chains. Change isoform for type VI collagen ?. Also, change in the legend of Figure 8.
Fig.7 What is the age of the mice shown?
DMD is not the only X-linked muscular dystrophy. If the authors refer to DMD they should use this term to avoid confusion, for example, FHL1 X-linked Emery Dreifuss type muscular dystrophy.
In summary, the review is written with evident expertise in the topic of muscular dystrophy fibrosis and proteomics and gives very useful information but I suggest revising it for clarity and focusing on the mdx 4cv mouse model and fibrosis and stating more clearly the criteria to select the information and data included in the review.
Author Response
Thanks for reviewing our manuscript [biomolecules-2469095] entitled ‘Extracellular matrix proteomics: the mdx-4cv mouse diaphragm as a surrogate for studying myofibrosis in dystrophinopathy’. We appreciate the time taken by Reviewer 2 to critically examine our submission. We have carried out a detailed revision of our paper and have listed our responses to individual points made by Reviewer 2 in below point-by-point responses. A copy of the revised manuscript with highlighted changes has been uploaded to illustrate the changes made in the R1 version of this paper.
Reviewer 2, Comment 1: ‘In this review, the authors make a considerable effort to present the components of the skeletal muscle matrisome at different anatomical levels (epimysium, perimysium, endomysium, MTJ, tendon etc) based on their studies and existing databases and literature (for example the widely used Matrisome project from MIT). The review starts with a useful and detailed technical overview of the different proteomic-based approaches to extract and analyse the extracellular matrix proteins. The role of the extracellular matrix in muscular dystrophies is paramount and so the information is very relevant’.
Response: We would like to thank Reviewer 2 for the positive evaluation of our review article. Our responses to specific points made by Reviewer 2 have been outlined in detail below.
Reviewer 2, Comment 2: ‘However, the information presented in the text, figures and tables is sometimes disconnected and the scope of the MS and the criteria to select the information in the review are not clear and appear a little random. For example, in section 3 the authors select a set of 25 proteins and list them in Table 1. According to the legend, these are the major proteins that they have identified in a previous study in the control mouse diaphragm. However, in the text right after the table, the authors describe the major families of extracellular matrix components without an apparent relationship with Table 1’.
Response: The MS data presented in Table 1 are shown as an example of a typical cohort of matrisomal proteins that can be routinely identified by the proteomic analysis of crude muscle specimens. In response to the Reviewer’s comment, we have revised the text following this table to better correlate the table to the general description of families of ECM proteins. Reference to Table 1 has been introduced in the revised section following Table 1 as follows:
Revised lines on Pages 10-11 (as highlighted in the R1 version of our paper):
‘Listed in Table 1 as a key component of the basal lamina is the heterotrimeric glycoprotein named laminin-211 …’.
‘Additional basal lamina enriched extracellular matrix proteins were detected as collagen isoforms IV (COL4A1), XV (COL15A1) and XVIII (COL18A1) [147], perlecan (HSPG-2) [148] and the nidogens named entactin (NID1) [149] and osteonidogen (NID2) [150], as listed in Table 1’.
‘Endomysium enriched extracellular matrix proteins are represented by specific collagens, as listed in Table 1, including collagen V (Col5a1) and collagen VI (Col6a1, Col6a2, Col6a5, Col6a6) …’.
‘As listed in Table 1, mass spectrometry detected key members of the family of non-structural matricellular proteins, including periostin (POSTN), thrombospondin (THBS1, THBS4) and fibulin isoform FBLN5 …’.
‘… transmission and signaling mechanisms. See Table 1 for the proteomic identification of these sarcolemmal protein receptors. The alpha-7/beta-1 integrins form …’.
Reviewer 2, Comment 3: ‘The order in which proteins are listed in Table 1 should be mentioned. Are they listed according to the number of peptides? Coverage?. Why were these proteins selected and not others? ‘.
Response: The protein hits included in Table 1 are proteomic hits with a peptide number of 2 and above for sequence recognition. Proteins were selected on high confidence amino acid sequence coverage. This information has been added to the revised manuscript. Importantly, it is mentioned in the text that the proteins listed in Table 1 cover proteoforms that are found throughout the endomysium, perimysium and epimysium, as well as protein species that are enriched in the basal lamina, myotendinous junctions, tendons and cartilage. Proteins are listed according to their location in the various layers of the matrisome.
Revised text below Table 1: * … and filter-aided sample preparation FASP method [86,87]. Proteins were selected on high confidence amino acid sequence coverage with a peptide number of 2 and above for sequence recognition. Listed are the accession number of detected proteins, the names of …’.
Reviewer 2, Comment 4: ‘I suggest focusing that section on describing the proteins in Table 1 if that is the scope of the review (diaphragm)’.
Response: We agree and the section below Table 1 describes the various ECM components, as already outlined above in response to Comment 2 of Reviewer 2.
Reviewer 2, Comment 5: ‘Similarly for Figure 4. Which criteria were used to select the proteins to be represented (what is the source)? What is the meaning of the spatial distribution of the proteins in the boxes? This refers to the composition of skeletal muscle in general and not just the diaphragm’.
Response: This figure correlates the approximate distribution of various ECM components to the various layers of the matrisome, as listed in Table 1, and is based on diaphragm data. However, this is a general and illustrative figure in our review to give an overview of the main components of the matrisome that can be routinely identified by MS-based proteomics of crude muscle extracts. We have introduced a new Figure 9 that summarizes changes in the abundance of ECM versus DGC proteins during fibrosis, and this scheme relates to Figure 4.
Reviewer 2, Comment 6: ‘Regarding the STRING network in Figure 5. Which list of proteins was selected to construct the network? There are only 25 proteins in Table 1 and in figure5 there are more than 25 nodes (proteins). What settings were used to construct the network? STRING allows for different parameters to build the edges depending on the type of interaction and level of confidence and this should be stated. Also, other descriptors such as the enrichment index (number of interactions or edges, PPI enrichment, could be used). The figure is very condensed and small and it is difficult to visualize the different clusters pointed out in the text by the authors. Perhaps unsupervised clustering analysis could be performed’.
Response: To improve the quality of Figure 5, the font of the gene (protein) names were enlarged and the display of network edges was changed from ‘evidence’, which gives considerable background, to ‘confidence’ with simplified line thickness indicating the strength of data support. The STRING network analysis was carried out with a list of 55 query items that represent routinely identified extracellular matrix protein isoforms, as listed in Table 1. Thus, this search did not just include overarching protein family names, but 55 individual protein species (symbolized by their respective gene name). The search list consisted of the matrisomal proteins that are encoded by the genes Lama2, Lamb1, Lamc1, Col4a1, Col4a2, Col15a1, Col18a1, Hspg2, Nid1, Nid2, Col5a1, Col6a1, Col6a2, Col6a5, Col6a6, Prelp, Dpt, Vtn, Fn1, Fga, Fgb, Fgg, Mfap4, Fbn1, Aspn, Bgn, Dcn, Fmod, Lum, Ogn, Postn, Thbs1, Thbs4, Fbln5, Itga7, Itgb1, Dag1, Col12a1, Fermt2, Col1a1, Col1a2, Comp, Cilp, Cilp2, Tgfbi, Tgm2, Anxa2, Anxa6, Serpina1d, Serpina1b, Serpina1c, Serpina1e, Serpinh1, Serpina3k and Ctsb. Species was set to ‘Mus musculus’ with the following parameters: (i) full STRING network analysis where the edges indicate both functional and physical protein associations, (ii) full set of active interaction sources consisting of textmining, experiments, databases, co‑expression, neighborhood, fusion and co‑occurrence, (iii) network edges using confidence whereby line thickness indicates the strength of data support, and (iv) network display mode with interactive scalable vector graphic (svg). This information has been added to the figure legend of revised Figure 5.
Page 13; revised figure legend: ‘Figure 5. Bioinformatic STRING analysis of potential protein-protein interactions in the extra-cellular matrix of skeletal muscle. Matrisomal proteins that were identified by mass spectrome-try, as listed in above Table 1, were analyzed by the method of Szklarczyk et al. [178]. The search list consisted of the matrisomal proteins that are encoded by the genes Lama2, Lamb1, Lamc1, Col4a1, Col4a2, Col15a1, Col18a1, Hspg2, Nid1, Nid2, Col5a1, Col6a1, Col6a2, Col6a5, Col6a6, Prelp, Dpt, Vtn, Fn1, Fga, Fgb, Fgg, Mfap4, Fbn1, Aspn, Bgn, Dcn, Fmod, Lum, Ogn, Postn, Thbs1, Thbs4, Fbln5, Itga7, Itgb1, Dag1, Col12a1, Fermt2, Col1a1, Col1a2, Comp, Cilp, Cilp2, Tgfbi, Tgm2, Anxa2, Anxa6, Serpina1d, Serpina1b, Serpina1c, Serpina1e, Serpinh1, Serpina3k and Ctsb. Species was set to ‘Mus musculus’ with the following parameters: (i) full STRING network anal-ysis where the edges indicate both functional and physical protein associations, (ii) full set of ac-tive interaction sources consisting of textmining, experiments, databases, co‑expression, neigh-borhood, fusion and co‑occurrence, (iii) network edges using confidence whereby line thickness indicates the strength of data support, and (iv) network display mode with interactive scalable vector graphic. Abbreviations used: ECM, extracellular matrix’.
Reviewer 2, Comment 7: ‘In section 4 the authors focus on the relevance of the mdx-4cv mouse model in the study of DMD. The purpose of listing experimental therapies in this mouse model is not clear and seems out of context. Given the aim of the review, I suggest focusing on the list of proteomic studies performed in this mouse model in skeletal muscle and biofluids. The data on other tissues do not seem to provide additional information’.
Response: In response to this comment, we would like to stress firstly the importance of being able to employ mouse models of Duchenne muscular dystrophy for the initial testing of the suitability of novel therapeutic approaches and secondly the clinical significance of the multi-systems etiology of dystrophinopathy. We therefore believe that it is crucial to list the studies that have evaluated experimental therapies using the mdx-4cv model in order to show that this genetic model is suitable for preclinical studies in the field of muscular dystrophy research. The listing of investigations with non-skeletal muscle tissues is in our opinion also of importance. We have changed the number sub-titles in Table 3 which now include ‘Proteomic profiling of skeletal muscle tissue specimens from the mdx-4cv model’ and ‘Proteomic profiling of non-skeletal muscle tissue specimens from the mdx-4cv model’ to make the distinction of the different analyses clearer. As recently discussed in a comprehensive review by our laboratories on the complexity of Duchenne muscular dystrophy (Reference [19] in this review: Ohlendieck, K.; Swandulla, D. Complexity of skeletal muscle degeneration: multi-systems pathophysiology and organ crosstalk in dystrophinopathy. Pflugers Arch. 2021, 473:1813-1839), dystrophinopathies can be considered multi-systems disorders. Although Duchenne muscular dystrophy is classified as a primary skeletal muscle disease, patients also show serious abnormalities in the cardio-respiratory system, the central nervous system, the liver, the GI-tract, the kidneys, bladder and the immune system. We therefore believe that the information on changes in non-skeletal muscle tissues is of biomedical importance and has relevance to the characterization of the suitability of the mdx-4cv model to investigate organ crosstalk and the multi-systems pathology of Duchenne muscular dystrophy.
Reviewer 2, Comment 8: ‘In section 4.3 there are some parts where the authors change from describing the fibrosis in mdx-4cv to then talking in general about changes detected by western blot in other models and DMD and I find this confusing’.
Response: To address this comment and avoid confusion on the various aspects discussed in Section 4, the sub-sections have been changed and now include: 4.1. Duchenne muscular dystrophy and fibrosis, 4.2. Characterization of the mdx-4cv mouse within the context of other dystrophic models, 4.3. Histological and biochemical characterization of fibrosis in the mdx-4cv mouse diaphragm, and 4.4. Proteomic profiling of fibrosis in the mdx-4cv mouse diaphragm. The cell biological studies and the immunoblotting are now discussed in a separate section, new sub-section 4.3. (Histological and biochemical characterization of fibrosis in the mdx-4cv mouse diaphragm).
Reviewer 2, Comment 9: ‘The criteria for the selection of proteins in Figure 8 and the list at the end of the results section are not clear. Are those the most differentially abundant proteins identified between wt and mdx-4cv diaphragms?’.
Response: The list of proteins are representative examples of frequently changed proteins in dystrophic mdx-4cv muscle tissues as compared to wild type. See for example the proteomic studies and reviews of the dystrophic diaphragm that are listed as reference [37] (Holland et al. Pathoproteomic profiling of the skeletal muscle matrisome in dystrophinopathy associated myofibrosis. Proteomics. 2016, 16:345-366, reference [55] (Gargan et al. Proteomic Identification of Markers of Membrane Repair, Regeneration and Fibrosis in the Aged and Dystrophic Diaphragm. Life (Basel). 2022, 12:1679) and reference [248] (Holland et al. Label-free mass spectrometric analysis of the mdx-4cv diaphragm identifies the matricellular protein periostin as a potential factor involved in dystrophinopathy-related fibrosis. Proteomics 2015, 15, 2318-2331).
The sentence before the listing of potential fibrosis markers has been changed to incorporate these 2 additional references as follows: ‘The following proteomic markers [37,55,284] could therefore be useful for the future establishment of an improved biomarker signature of dystrophinopathy-associated myofibrosis: …’
Reviewer 2, Comment 10: ‘Minor points; Section 4.3 COL-VI isoform: Collagen VI is a collagen type made of alpha-chains. Change isoform for type VI collagen ?. Also, change in the legend of Figure 8’.
Response: This point has been addressed in the revised text and figure legend of Figure 8.
Line 673: ‘… illustrate the elevated levels of collagen isoform COL-VI, which in muscle tissue consists of 3 alpha chains [158], the annexin isoform …’.
Figure legend of Figure 8: ‘… and loading control, alpha chain-containing collagen VI (COL-VI) of the endomysium as an abundant marker of the extracellular matrix, the matricellular protein …’.
Reviewer 2, Comment 11: ‘Fig.7 What is the age of the mice shown?’.
Response: The missing information on the age of the mice (12-month old) was added to revised Figure 7.
Figure legend of Figure 7: ‘Histological and immunofluorescence microscopical analysis of dystrophic mdx-4cv mouse diaphragm muscle. Shown are transverse sections of 12-month old diaphragm muscles from wild type (wt) versus age-matched dystrophic mdx-4cv mice. Cryosections were stained with …’.
Reviewer 2, Comment 12: ‘DMD is not the only X-linked muscular dystrophy. If the authors refer to DMD they should use this term to avoid confusion, for example, FHL1 X-linked Emery Dreifuss type muscular dystrophy’.
Response: To address this point, the first sentence of the third paragraph in the Introduction section has been rephrased to clearly link the term ‘X-linked muscular dystrophy’ in this review to ‘primary abnormalities in the DMD gene’, as well as other parts in the text.
Line 65:’ X-linked muscular dystrophies that are based on primary abnormalities in the human DMD gene [2] and trigger the almost complete absence of the full-length Dp427-M isoform of the cytoskeletal protein dystrophin [18] are highly progressive muscle wasting diseases.’.
Line 466: ‘… and its involvement in X-linked Duchenne muscular dystrophy …’.
Line 614: ‘… novel therapeutic strategies to treat X-linked Duchenne muscular dystrophy …’.
Line 708: ‘… novel treatment strategies for X-linked Duchenne muscular dystrophy …’.
Line 741: ‘… including X-linked Duchenne muscular dystrophy …’.
Line 799: ‘… monitoring of X-linked Duchenne muscular dystrophy …’.
Reviewer 2, Comment 13: ‘In summary, the review is written with evident expertise in the topic of muscular dystrophy fibrosis and proteomics and gives very useful information but I suggest revising it for clarity and focusing on the mdx 4cv mouse model and fibrosis and stating more clearly the criteria to select the information and data included in the review.
Response: We would like to thank Reviewer 2 for the thorough review of our article and constructive criticism. We hope that the above outlined changes in the presentation of our manuscript have addressed the main points made by Reviewer 2 and that the paper is now acceptable for publication.
Reviewer 3 Report
The review article entitled " Extracellular matrix proteomics: the mdx-4cv mouse diaphragm as a surrogate for studyng myofibrosis in dystrophinopathy" by Dowling and colleagues is well conceived and gives and excellent state-of-the-art about the myofibrosis and, more in general, about the extracellular matrix adaptations in muscular dystrophy. The article also provide an accurate methodology for proteomic analysis.
Two points can be addressed to further improve the quality of the article:
1) mdx-4cv is the reference model of the article, some critical features of the model have been described in the text (mutation, low number of dystrophin-positive revertants fibers, etc..), however I think the article can benefit by a brief description of the disease progression of this model, such as the pick of the muscle degeneration that is around 3-4 weeks of age. Differences in the prognosis between mdx-23 and mdx-4cv should be highlighted in the text, as well as those in the comparative overview between DMD and mdx-4cv of figure 6.
2) Golden Retriever muscular dystrophy (GRMD), homologue of DMD, is a spontaneous, X-linked, fatal disease. Since its importance as alternative model, compared the most used murine ones, I suggest to include a brief description of this model in the text highlighting its pathophysiology, including muscle degeneration, cardiac fibrosis, etc ( doi: 10.3727/096368912X638919).
Author Response
Thanks for reviewing our manuscript [biomolecules-2469095] entitled ‘Extracellular matrix proteomics: the mdx-4cv mouse diaphragm as a surrogate for studying myofibrosis in dystrophinopathy’. We appreciate the time taken by Reviewer 3 to critically examine our submission. We have carried out a detailed revision of our paper and have listed our responses to individual points made by Reviewer 3 in below point-by-point responses. A copy of the revised manuscript with highlighted changes has been uploaded to illustrate the changes made in the R1 version of this paper.
Reviewer 3, Comment 1: ‘The review article entitled " Extracellular matrix proteomics: the mdx-4cv mouse diaphragm as a surrogate for studyng myofibrosis in dystrophinopathy" by Dowling and colleagues is well conceived and gives and excellent state-of-the-art about the myofibrosis and, more in general, about the extracellular matrix adaptations in muscular dystrophy. The article also provide an accurate methodology for proteomic analysis’.
Response: We would like to thank Reviewer 3 for the positive evaluation of our review article. Our responses to specific points made by Reviewer 3 have been outlined in detail below.
Reviewer 3, Comment 2: ‘Two points can be addressed to further improve the quality of the article: 1) mdx-4cv is the reference model of the article, some critical features of the model have been described in the text (mutation, low number of dystrophin-positive revertants fibers, etc..), however I think the article can benefit by a brief description of the disease progression of this model, such as the pick of the muscle degeneration that is around 3-4 weeks of age. Differences in the prognosis between mdx-23 and mdx-4cv should be highlighted in the text, as well as those in the comparative overview between DMD and mdx-4cv of figure 6’.
Response: We agree and have added a more detailed description of the disease process in the mdx-4cv mouse model of Duchenne muscular dystrophy in revised Section 4.2. To address the point on the direct comparison of the mdx-23 model versus the mdx-4cv model, which has also been raised by another reviewer, we have added this information to new Table 2 and in revised Figure 6.
Since this point was also raised by another reviewer, the comparison of the mdx-4cv mouse to other genetic mouse models of dystrophinopathy was carried out which includes additional text and a new table, Table 2, as well as new references that focus on mouse models of muscular dystrophy. The new text section and new Table 2 have been introduced in Section 4.2 (Characterization of the mdx-4cv mouse within the context of other dystrophic models) on Pages 15-17 in the revised manuscript summarizing the different types of mouse models of DMD. The other tables and references have been renumbered accordingly.
This section also contains the more detailed description of the disease process in the mdx-4cv mouse model and its comparison to the mdx-23 model.
Revised text on Pages 15-19 (including new Table 2; please see revised R1 version):
‘4.2. Characterization of the mdx-4cv mouse within the context of other dystrophic models
Both spontaneous and bioengineered mouse models that lack the dystrophin protein isoform Dp427-M have been instrumental for the detailed elucidation of the molecular and cellular pathogenesis of Duchenne muscular, as well as the evaluation of experimental treatment strategies [210-212]. A comparative listing of major genetic mouse models of dystrophinopathy is provided in Table 2. This includes the original mdx-23 mouse as compared to a variety of modified models including mdx/Dtna, mdx/Cmah, mdx/Utr, mdx/alpha7, mdx/Myod1, mdx-2cv, mdx-3cv, mdx-4cv, mdx-4cv/mTR-G2, mdx-5cv, mdx-52, mdx-beta-geo, Dmd-null, hDMD/mdx-45 and hDMD/mdx-52. The table lists information on the genetic abnormalities that characterize the various mouse models and the observed se-verity of the dystrophic phenotype in individual models of Duchenne muscular dystrophy. Importantly, Table 2 gives an overview of the effect of mutations on the expression of distinct dystrophin isoforms ranging from Dp71 to Dp427. Depending on the genetic abnormality within the Dmd gene, impaired expression patterns might only cause the loss of the full-length dystrophins Dp427-M, Dp427-B and Dp427-P, or also additional effects on the tissue-specific density of the shorter dystrophins Dp260-R, Dp140-B/K, Dp116-S, Dp71 and/or Dp45. Detailed comparisons of dystrophic mouse models have been carried out in recent reviews [213-215].
Compared to the onset of dystrophic changes in the skeletal musculature at ap-proximately 1-month of age in the mdx-23 mouse and a maximum lifespan of approxi-mately 2 years, the modified mdx-type mice differ considerably in the onset of histopathological changes, ranging from 2-8 weeks, and their longevity, ranging in lifespan from 1-23 months [216]. Both the spontaneously mutated mdx-23 mouse and the chemically mutated mdx-4cv mouse have comparable survival rates and similar onsets of moderate hindlimb degeneration, as judged by histopathology, at 3-4 weeks of age in association with physiological dysfunction such as reduced muscle force. The mdx-23 mouse is characterized by extensive cycles of degeneration and regeneration from 3-8 weeks, illustrated by high levels of central myonucleation, and some stabilization with age [38-40]. The mdx-4cv musculature shows large variation in skeletal muscle fiber sizes. In both model systems, the dystrophin-deficient diaphragm is severely affected starting at 3-4 weeks of age and shows high levels of reactive myofibrosis during aging [213-215]. In analogy to the skeletal musculature, the reduced presence of the cardiac dystro-phin-glycoprotein complex causes sarcolemmal disintegration, myonecrosis, fatty tissue replacement, fibrotic scarring and interstitial inflammation in the mdx-type heart [217]. The onset of cardiomyopathy is at approximately 6-9 months of age and associated with considerable proteome-wide changes during aging [218].
…
Table 2: Comparison of major genetic mouse models of Duchenne muscular dystrophy*.
…
Skeletal muscle fibrosis is seen in different mdx-type muscles to varying degrees [46-48,226], with the diaphragm being the most structurally affected and fibrotic muscle tissue [49-52,227,228]. However, one of the most widely employed dystrophic animal models, the mdx-23 mouse [38], exhibits a relatively high frequency of revertant fibers [39] and a relatively mild dystrophic phenotype [40]. In contrast, the chemically mutated mdx-4cv mouse which harbors a nonsense mutation in exon-53 [41-43] has a drastically reduced number of dystrophin-positive revertants [45]. Figure 6 provides a comparison of the main genetic, biochemical, physiological and histological features of muscular ab-normalities in Duchenne muscular dystrophy versus the widely used mdx-23 and mdx-4cv mouse models of dystrophinopathy’.
Reviewer 3, Comment 3: ‘2) Golden Retriever muscular dystrophy (GRMD), homologue of DMD, is a spontaneous, X-linked, fatal disease. Since its importance as alternative model, compared the most used murine ones, I suggest to include a brief description of this model in the text highlighting its pathophysiology, including muscle degeneration, cardiac fibrosis, etc ( doi: 10.3727/096368912X638919)’.
Response: We agree and have added additional information on larger DMD models, such as the GRMD, and quoted relevant papers in revised Section 4.2., including the suggested article by Cassano et al. (Cassano M, Berardi E, Crippa S, Toelen J, Barthelemy I, Micheletti R, Chuah M, Vandendriessche T, Debyser Z, Blot S, Sampaolesi M. Alteration of cardiac progenitor cell potency in GRMD dogs. Cell Transplant. 2012;21(9):1945-67).
Revised text on Pages 15-16 (last paragraph of revised Section 4.2. Characterization of the mdx-4cv mouse within the context of other dystrophic models)
‘… In addition to the most frequently used murine models with their rapid disease progression, large animal models of Duchenne muscular dystrophy exist in the form of canine and porcine mutants [219-221]. The Golden Retriever muscular dystrophy (GRMD) model exhibits a splice site mutation in intron-6, which causes skipping of exon-7 and a resulting out-of-frame DMD transcript in exon-8 [220]. The complexity and progression of the clinical syndrome is severe in the GRMD model including cardiomyopathic complications making dystrophic dogs good model systems for translational studies [222,223]. The porcine model of dystrophinopathy (DMD pig) is characterized by a deletion in Exon-52 of the DMD gene and exhibits progressive cardiomyopathy [224]. In conjunction with mouse models, larger animals can be utilized to study pathophysiological mechanisms, validate the suitability and robustness of novel biomarker candidates, and be helpful during the preclinical testing phase of new therapeutic approaches such as gene editing [219,221]. Large disease model animals are highly useful for facilitating the efficient translation of novel diagnostic methods to the clinical setting, such as imaging technology for the monitoring of myofibrosis [225].
Reference [222] Cassano, M.; Berardi, E.; Crippa, S.; Toelen, J.; Barthelemy, I.; Micheletti, R.; Chuah, M.; Vandendriessche, T.; Debyser, Z.; Blot, S.; Sampaolesi, M. Alteration of cardiac progenitor cell potency in GRMD dogs. Cell Transplant. 2012, 21, 1945-1967.
Round 2
Reviewer 2 Report
Thank you for your responses to the comments which I hope have helped to finalise the MS for publication